# Learning to Control the Smoothness of GCN Features

## Abstract

The pioneering work of Oono & Suzuki [ICLR, 2020] and Cai & Wang [arXiv:2006.13318] analyze the smoothness of graph convolutional network (GCN) features. Their results reveal an intricate empirical correlation between node classification accuracy and the ratio of smooth to non-smooth feature components. However, the optimal ratio that favors node classification is unknown, and the non-smooth features of deep GCN with ReLU or leaky ReLU activation function diminish. In this paper, we propose a new strategy to let GCN learn node features with a desired smoothness to enhance node classification. Our approach has three key steps: (1) We establish a geometric relationship between the input and output of ReLU or leaky ReLU. (2) Building on our geometric insights, we augment the message-passing process of graph convolutional layers (GCLs) with a learnable term to modulate the smoothness of node features with computational efficiency. (3) We investigate the achievable ratio between smooth and non-smooth feature components for GCNs with the augmented message passing scheme. Our extensive numerical results show that the augmented message passing remarkably improves node classification for GCN and some related models.

## 1 Introduction

Let $G = (V, E)$ be an undirected graph with $V = \{v_i\}_{i=1}^n$ and $E$ be the set of nodes and edges, resp. Let $\boldsymbol{A} \in \mathbb{R}^{n \times n}$ be the adjacency matrix of the graph with $A_{ij} = \mathbf{1}_{(i,j) \in E}$, where $\mathbf{1}$ is the indicator function. Furthermore, let $\boldsymbol{G}$ be the following (augmented) normalized adjacency matrix

$$\boldsymbol{G} := (\boldsymbol{D} + \boldsymbol{I})^{-\frac{1}{2}} (\boldsymbol{I} + \boldsymbol{A})(\boldsymbol{D} + \boldsymbol{I})^{-\frac{1}{2}} = \tilde{\boldsymbol{D}}^{-\frac{1}{2}} \tilde{\boldsymbol{A}} \tilde{\boldsymbol{D}}^{-\frac{1}{2}}, \tag{1}$$

where $\boldsymbol{I}$ is the identity matrix, $\boldsymbol{D}$ is the degree matrix with $D_{ii} = \sum_{j=1}^n A_{ij}$, and $\tilde{\boldsymbol{A}} := \boldsymbol{A} + \boldsymbol{I}$ and $\tilde{\boldsymbol{D}} := \boldsymbol{D} + \boldsymbol{I}$. Starting from the initial node features $\boldsymbol{H}^0 := [(\boldsymbol{h}_1^0)^\top, \ldots, (\boldsymbol{h}_n^0)^\top]^\top \in \mathbb{R}^{d \times n}$ with $\boldsymbol{h}_i^0 \in \mathbb{R}^d$ being the $i^{th}$ node feature vector, the graph convolutional network (GCN) [20] learns node representations using the following graph convolutional layer (GCL) transformation

$$\boldsymbol{H}^l = \sigma(\boldsymbol{W}^l \boldsymbol{H}^{l-1} \boldsymbol{G}), \tag{2}$$

where $\sigma$ is the activation function, e.g. ReLU [25], and $\boldsymbol{W}^l \in \mathbb{R}^{d \times d}$ is learnable. GCL smooths feature vectors of the neighboring nodes. The smoothness of features helps node classification; see e.g. [22, 31, 5], resonating with the idea of classical semi-supervised learning approaches [41, 38]. Accurate node classification requires a balance between smooth and non-smooth components of GCN features [27]. Besides graph convolutional networks (GCNs) stacking GCLs, many other graph neural networks (GNNs) have been developed using different mechanisms, including spectral methods [3, 9], spatial methods [12, 30], sampling methods [13, 36], and the attention mechanism [30]. Many other GNN models can be found in recent surveys or monographs; see, e.g. [15, 1, 33, 39, 14].

Deep neural networks usually outperform shallow architectures, and a remarkable example is convolutional neural networks [21, 16]. However, this does not carry to GCNs; deep GCNs tend to perform

significantly worse than shallow models [5]. In particular, the node feature vectors learned by deep GCNs tend to be identical over each connected component of the graph; this phenomenon is referred to as ***over-smoothing*** [22, 26, 27, 4, 5, 32], which not only occurs for GCN but also for many other GNNs, e.g., GraphSage [13] and MPNN [12]. Intuitively, each GCL smooths neighboring node features, benefiting node classification [22, 31, 5]. However, stacking these smoothing layers will inevitably homogenize node features. Algorithms have been developed to alleviate the over-smoothing issue of GNNs, including decoupling prediction and message passing [11], skip connection and batch normalization [18, 7, 6], graph sparsification [29], jumping knowledge [34], scattering transform [24], PairNorm [37], and controlling the Dirichlet energy of node features [40].

From a theoretical perspective, it is proved that deep GCNs using ReLU or leaky ReLU activation function learn homogeneous node features [27, 4]. In particular, [27] shows that the distance of node features to the eigenspace $\mathcal{M}$ – corresponding to the largest eigenvalue 1 of matrix $\boldsymbol{G}$ in (1) – goes to zero when the depth of GCN with ReLU goes to infinity. Meanwhile, [27] empirically studies the intricate correlation between node classification accuracy and the ratio between smooth and non-smooth components of GCN node features, i.e., projections of node features onto eigenspace $\mathcal{M}$ and its orthogonal complement $\mathcal{M}^\perp$, resp. The empirical results of [27] indicate that ***both smooth and non-smooth components of node features are crucial for accurate node classification***, while the ratio between smooth and non-smooth components to achieve optimal accuracy is unknown and task-dependent. Furthermore, [4] proves that the Dirichlet energy – another smoothness measure for node features – goes to zero when the depth of GCN with ReLU or leaky ReLU goes to infinity.

A crucial step in the proofs of [27, 4] is that ReLU and leaky ReLU reduce the distance of feature vectors to $\mathcal{M}$ and their Dirichlet energy. However, [4] points out that ***over-smoothing – characterized by the distance of features to eigenspace $\mathcal{M}$ or the Dirichlet energy – is a misnomer***; the real smoothness should be characterized by a ***normalized smoothness***, e.g., normalizing the Dirichlet energy by the magnitude of the features. ***The ratio between smooth and non-smooth components of node features – studied in [27] – is closely related to the normalized smoothness***. Nevertheless, analyzing the normalized smoothness of node features learned by GCN with ReLU or leaky ReLU remains an open problem [4]. Moreover, it is interesting to ask if analyzing the normalized smoothness can result in any new understanding of GCN features and algorithms to improve GCN's performance.

## 1.1 Our contribution

We aim to (1) establish a new geometric understanding of how GCL smooths GCN features and (2) develop an efficient algorithm to let GCN and related models learn node features with a desired normalized smoothness to improve node classification. We summarize our main contributions towards achieving our goal as follows:

- We prove that there is a high-dimensional sphere underlying the input and output vectors of ReLU or leaky ReLU. This geometric characterization not only implies theories in [27, 4] but also informs that adjusting the projection of input onto eigenspace $\mathcal{M}$ can alter the smoothness of the output vectors. See Section 3 for details.

- We show that both ReLU and leaky ReLU reduce the distance of node features to eigenspace $\mathcal{M}$, i.e., ReLU and leaky ReLU smooth their input vectors without considering their magnitude. In contrast, when taking the magnitude into account, ReLU and leaky ReLU can increase, decrease, or preserve the normalized smoothness of each dimension of the input vectors; see Sections 3 and 4.

- Inspired by our established geometric relationship between the input and output of ReLU or leaky ReLU, we study how adjusting the projection of input onto eigenspace $\mathcal{M}$ affects both normalized and unnormalized smoothness of the output vectors. We show that the distance of the output to eigenspace $\mathcal{M}$ is no greater than that of the original input – no matter how we adjust the input by changing its projection onto $\mathcal{M}$. In contrast, adjusting the projection of input vectors onto $\mathcal{M}$ can change the normalized smoothness of output to any desired value; see details in Section 4.

- Based on our theory, we propose a computationally efficient smoothness control term (SCT) to let GCN and related models learn node features with a desired (normalized) smoothness to improve node classification. We comprehensively validate the benefits of our proposed SCT in improving node classification – for both homophilic and heterophilic graphs – using a few of the most representative GCN-style models. See Sections 5 and 6 for details.

As far as we know, our work is the first thorough study of how ReLU and leaky ReLU affect the smoothness of node features both with and without considering their magnitude.

## 1.2 Additional related works

Controlling the smoothness of node features to improve the performance of GCNs is another line of related work. For instance, [37] designs a normalization layer to prevent node features from becoming too similar to each other, and [40] constrains the Dirichlet energy to control the smoothness of node features without considering the effects of nonlinear activation functions. While there has been effort in understanding and alleviating the over-smoothing of GCNs and controlling the smoothness of node features, there is a shortage of theoretical examination of how activation functions affect the smoothness of node features, specifically accounting for the magnitude of features.

## 1.3 Notation and Organization

**Notation.** We denote the $\ell_2$-norm of a vector $\boldsymbol{u}$ as $\|\boldsymbol{u}\|$. For vectors $\boldsymbol{u}$ and $\boldsymbol{v}$, we use $\langle \boldsymbol{u}, \boldsymbol{v} \rangle$, $\boldsymbol{u} \odot \boldsymbol{v}$, and $\boldsymbol{u} \otimes \boldsymbol{v}$ to denote their inner, Hadamard, and Kronecker product, resp. For a matrix $\boldsymbol{A}$, we denote its $(i,j)^{th}$ entry, transpose, and inverse as $A_{ij}$, $\boldsymbol{A}^\top$, and $\boldsymbol{A}^{-1}$, resp. We denote the trace of $\boldsymbol{A} \in \mathbb{R}^{n \times n}$ as $\mathrm{Trace}(\boldsymbol{A}) = \sum_{i=1}^{n} A_{ii}$. For two matrices $\boldsymbol{A}$ and $\boldsymbol{B}$, we denote the Frobenius inner product as $\langle \boldsymbol{A}, \boldsymbol{B} \rangle_F := \mathrm{Trace}(\boldsymbol{A}\boldsymbol{B}^\top)$ and the Frobenius norm of $\boldsymbol{A}$ as $\|\boldsymbol{A}\|_F := \sqrt{\langle \boldsymbol{A}, \boldsymbol{A} \rangle}$.

**Organization.** We provide preliminaries in Section 2. In Section 3, we establish a geometric characterization of how ReLU and leaky ReLU affect the smoothness of their input vectors. We study the smoothness of each dimension of node features and take their magnitude into account in Section 4. Our proposed SCT is presented in Section 5. We comprehensively verify the efficacy of the proposed SCT in Section 6. Technical proofs and more experimental results are provided in the appendix.

## 2 Preliminaries and Existing Results

From the spectral graph theory [8], we can sort eigenvalues of matrix $\boldsymbol{G}$ in (1) as $1 = \lambda_1 = \ldots = \lambda_m > \lambda_{m+1} \geq \ldots \geq \lambda_n > -1$, where $m$ is the number of connected components of the graph. We decompose $V = \{v_k\}_{k=1}^{n}$ into $m$ connected components $V_1, \ldots, V_m$. Let $\boldsymbol{u}_i = (\mathbf{1}_{\{v_k \in V_i\}})_{1 \leq k \leq n}$ be the indicator vector of $V_i$, i.e., the $k^{th}$ coordinate of $\boldsymbol{u}_i$ is one if the $k^{th}$ node $v_k$ lies in the connected component $V_i$; zero otherwise. Moreover, let $\boldsymbol{e}_i$ be the eigenvector associated with $\lambda_i$, then $\{\boldsymbol{e}_i\}_{i=1}^{n}$ forms an orthonormal basis of $\mathbb{R}^n$. Notice that $\{\boldsymbol{e}_i\}_{i=1}^{m}$ spans the eigenspace $\mathcal{M}$ – corresponding to eigenvalue 1 of matrix $\boldsymbol{G}$, and $\{\boldsymbol{e}_i\}_{i=m+1}^{n}$ spans the orthogonal complement of $\mathcal{M}$, denoted by $\mathcal{M}^\perp$. The paper [27] connects the indicator vectors $\boldsymbol{u}_i$s with the space $\mathcal{M}$. In particular, we have

**Proposition 2.1** ([27]). *All eigenvalues of matrix $\boldsymbol{G}$ lie in the interval $(-1, 1]$. Furthermore, the nonnegative vectors $\{\tilde{\boldsymbol{D}}^{\frac{1}{2}} \boldsymbol{u}_i / \|\tilde{\boldsymbol{D}}^{\frac{1}{2}} \boldsymbol{u}_i\|\}_{1 \leq i \leq m}$ form an orthonormal basis of $\mathcal{M}$.*

For any matrix $\boldsymbol{H} := [\boldsymbol{h}_1, \ldots, \boldsymbol{h}_n] \in \mathbb{R}^{d \times n}$, we have the decomposition $\boldsymbol{H} = \boldsymbol{H}_{\mathcal{M}} + \boldsymbol{H}_{\mathcal{M}^\perp}$ with $\boldsymbol{H}_{\mathcal{M}} = \sum_{i=1}^{m} \boldsymbol{H} \boldsymbol{e}_i \boldsymbol{e}_i^\top$ and $\boldsymbol{H}_{\mathcal{M}^\perp} = \sum_{i=m+1}^{n} \boldsymbol{H} \boldsymbol{e}_i \boldsymbol{e}_i^\top$ such that $\langle \boldsymbol{H}_{\mathcal{M}}, \boldsymbol{H}_{\mathcal{M}^\perp} \rangle_F = \mathrm{Trace}\left( \sum_{i=1}^{m} \boldsymbol{H} \boldsymbol{e}_i \boldsymbol{e}_i^\top (\sum_{j=m+1}^{n} \boldsymbol{H} \boldsymbol{e}_j \boldsymbol{e}_j^\top)^\top \right) = 0$, implying that $\|\boldsymbol{H}\|_F^2 = \|\boldsymbol{H}_{\mathcal{M}}\|_F^2 + \|\boldsymbol{H}_{\mathcal{M}^\perp}\|_F^2$.

### 2.1 Existing smoothness notions of node features

**Distance to the eigenspace $\mathcal{M}$.** Oono et al. [27] study the smoothness of features $\boldsymbol{H} := [\boldsymbol{h}_1, \ldots, \boldsymbol{h}_n]$ using their distance to the eigenspace $\mathcal{M}$ as an unnormalized smoothness notion.

**Definition 2.2** ([27]). Let $\mathbb{R}^d \otimes \mathcal{M}$ be the subspace of $\mathbb{R}^{d \times n}$ consisting of the sum $\sum_{i=1}^{m} \boldsymbol{w}_i \otimes \boldsymbol{e}_i$, where $\boldsymbol{w}_i \in \mathbb{R}^d$ and $\{\boldsymbol{e}_i\}_{i=1}^{m}$ is an orthonormal basis of the eigenspace $\mathcal{M}$. Then we define $\|\boldsymbol{H}\|_{\mathcal{M}^\perp}$ – the distance of node features $\boldsymbol{H}$ to the eigenspace $\mathcal{M}$ – as follows:

$$\|\boldsymbol{H}\|_{\mathcal{M}^\perp} := \inf_{\boldsymbol{Y} \in \mathbb{R}^d \otimes \mathcal{M}} \|\boldsymbol{H} - \boldsymbol{Y}\|_F = \Big\| \boldsymbol{H} - \sum_{i=1}^{m} \boldsymbol{H} \boldsymbol{e}_i \boldsymbol{e}_i^\top \Big\|_F.$$

With the decomposition $\boldsymbol{H} = \boldsymbol{H}_{\mathcal{M}} + \boldsymbol{H}_{\mathcal{M}^\perp}$, $\|\cdot\|_{\mathcal{M}^\perp}$ can be related to $\|\cdot\|_F$ as follows:

$$\|\boldsymbol{H}\|_{\mathcal{M}^\perp} = \|\boldsymbol{H} - \boldsymbol{H}_{\mathcal{M}}\|_F = \|\boldsymbol{H}_{\mathcal{M}^\perp}\|_F. \tag{3}$$

**Dirichlet energy.** The paper [4] studies the unnormalized smoothness of node features using Dirichlet energy, which is defined as follows:

**Definition 2.3** ([4]). Let $\tilde{\Delta} = \boldsymbol{I} - \boldsymbol{G}$ be the (augmented) normalized Laplacian, then the Dirichlet energy $\|\boldsymbol{H}\|_E$ of node features $\boldsymbol{H}$ is defined by $\|\boldsymbol{H}\|_E^2 := \mathrm{Trace}(\boldsymbol{H} \tilde{\Delta} \boldsymbol{H}^\top)$.

131 **Normalized Dirichlet energy.** [4] points out that the real smoothness of node features $\boldsymbol{H}$ should be
132 measured by the normalized Dirichlet energy $\mathrm{Trace}(\boldsymbol{H}\tilde{\Delta}\boldsymbol{H}^\top)/\|\boldsymbol{H}\|_F^2$. This normalized measurement
133 is essential because data often originates from various sources with diverse measurement units or
134 scales. By normalization, we can mitigate biases resulting from these different scales.

## 2.2 Two existing theories of over-smoothing

136 Let $\lambda = \max\{|\lambda_i| \mid \lambda_i < 1\}$ be the second largest magnitude of $\boldsymbol{G}$'s eigenvalues, and $s_l$ be the largest
137 singular value of weight matrix $\boldsymbol{W}^l$. [27] shows that $\|\boldsymbol{H}^l\|_{\mathcal{M}^\perp} \leq s_l\lambda\|\boldsymbol{H}^{l-1}\|_{\mathcal{M}^\perp}$ under GCL when
138 $\sigma$ is ReLU. Therefore, $\|\boldsymbol{H}^l\|_{\mathcal{M}^\perp} \to 0$ as $l \to \infty$ if $s_l\lambda < 1$, indicating node features converge to $\mathcal{M}$
139 and results in over-smoothing. A crucial step in the analysis in [27] is that $\|\sigma(\boldsymbol{Z})\|_{\mathcal{M}^\perp} \leq \|\boldsymbol{Z}\|_{\mathcal{M}^\perp}$, for
140 any matrix $\boldsymbol{Z}$ when $\sigma$ is ReLU, i.e., ReLU reduces the distance to $\mathcal{M}$. [27] points out that it is hard
141 to extend the above result to other activation functions even leaky ReLU.

142 Instead of considering $\|\boldsymbol{H}\|_{\mathcal{M}^\perp}$, [4] shows that $\|\boldsymbol{H}^l\|_E \leq s_l\lambda\|\boldsymbol{H}^{l-1}\|_E$ under GCL when $\sigma$ is
143 ReLU or leaky ReLU. Hence, $\|\boldsymbol{H}^l\|_E \to 0$ as $l \to \infty$, implying over-smoothing of GCNs. Note that
144 $\|\boldsymbol{H}\|_{\mathcal{M}^\perp} = 0$ or $\|\boldsymbol{H}^l\|_E = 0$ indicates homogeneous node features. The proof in [4] applies to GCN
145 with both ReLU and leaky ReLU by establishing the inequality $\|\sigma(\boldsymbol{Z})\|_E \leq \|\boldsymbol{Z}\|_E$ for any matrix $\boldsymbol{Z}$.

# 3 Effects of Activation Functions: A Geometric Characterization

147 In this section, we present a geometric relationship between the input and output vectors of ReLU or
148 leaky ReLU. We use $\|\boldsymbol{H}\|_{\mathcal{M}^\perp}$ as the unnormalized smoothness notion for all subsequent analyses
149 since we observe that $\|\boldsymbol{H}\|_{\mathcal{M}^\perp}$ and $\|\boldsymbol{H}\|_E$ are equivalent as seminorms. In particular, we have

150 **Proposition 3.1.** $\|\boldsymbol{H}\|_{\mathcal{M}^\perp}$ *and* $\|\boldsymbol{H}\|_E$ *are two equivalent seminorms, i.e., there exist two constants*
151 $\alpha, \beta > 0$ *s.t.* $\alpha\|\boldsymbol{H}\|_{\mathcal{M}^\perp} \leq \|\boldsymbol{H}\|_E \leq \beta\|\boldsymbol{H}\|_{\mathcal{M}^\perp}$, *for any* $\boldsymbol{H} \in \mathbb{R}^{d \times n}$.

## 3.1 ReLU

153 Let $\sigma(x) = \max\{x, 0\}$ be ReLU. The first main result of this paper is that there is a high-dimensional
154 sphere underlying the input and output of ReLU; more precisely, we have

**Proposition 3.2** (ReLU). *For any* $\boldsymbol{Z} = \boldsymbol{Z}_{\mathcal{M}} + \boldsymbol{Z}_{\mathcal{M}^\perp} \in \mathbb{R}^{d \times n}$, *let* $\boldsymbol{H} = \sigma(\boldsymbol{Z}) = \boldsymbol{H}_{\mathcal{M}} + \boldsymbol{H}_{\mathcal{M}^\perp}$.
*Then* $\boldsymbol{H}_{\mathcal{M}^\perp}$ *lies on the high-dimensional sphere centered at* $\boldsymbol{Z}_{\mathcal{M}^\perp}/2$ *with radius*

$$r := \left(\|\boldsymbol{Z}_{\mathcal{M}^\perp}/2\|_F^2 - \langle \boldsymbol{H}_{\mathcal{M}}, \boldsymbol{H}_{\mathcal{M}} - \boldsymbol{Z}_{\mathcal{M}}\rangle_F\right)^{1/2}.$$

155 *In particular,* $\boldsymbol{H}_{\mathcal{M}^\perp}$ *lies inside the ball centered at* $\boldsymbol{Z}_{\mathcal{M}^\perp}/2$ *with radius* $\|\boldsymbol{Z}_{\mathcal{M}^\perp}/2\|_F$ *and hence we*
156 *have* $\|\boldsymbol{H}\|_{\mathcal{M}^\perp} \leq \|\boldsymbol{Z}\|_{\mathcal{M}^\perp}$.

## 3.2 Leaky ReLU

158 Now we consider leaky ReLU $\sigma_a(x) = \max\{x, ax\}$, where $0 < a < 1$ is a positive scalar. Similar
159 to ReLU, we have the following result for leaky ReLU

**Proposition 3.3** (Leaky ReLU). *For any* $\boldsymbol{Z} = \boldsymbol{Z}_{\mathcal{M}} + \boldsymbol{Z}_{\mathcal{M}^\perp} \in \mathbb{R}^{d \times n}$, *let* $\boldsymbol{H} = \sigma_a(\boldsymbol{Z}) = \boldsymbol{H}_{\mathcal{M}} + \boldsymbol{H}_{\mathcal{M}^\perp}$. *Then* $\boldsymbol{H}_{\mathcal{M}^\perp}$ *lies on the high-dimensional sphere centered at* $(1 + a)\boldsymbol{Z}_{\mathcal{M}^\perp}/2$ *with radius*

$$r_a := \left(\|(1 - a)\boldsymbol{Z}_{\mathcal{M}^\perp}/2\|_F^2 - \langle \boldsymbol{H}_{\mathcal{M}} - \boldsymbol{Z}_{\mathcal{M}}, \boldsymbol{H}_{\mathcal{M}} - a\boldsymbol{Z}_{\mathcal{M}}\rangle_F\right)^{1/2}.$$

160 *In particular,* $\boldsymbol{H}_{\mathcal{M}^\perp}$ *lies inside the ball centered at* $(1 + a)\boldsymbol{Z}_{\mathcal{M}^\perp}/2$ *with radius* $\|(1 - a)\boldsymbol{Z}_{\mathcal{M}^\perp}/2\|_F$
161 *and hence we see that* $a\|\boldsymbol{Z}\|_{\mathcal{M}^\perp} \leq \|\boldsymbol{H}\|_{\mathcal{M}^\perp} \leq \|\boldsymbol{Z}\|_{\mathcal{M}^\perp}$.

## 3.3 Implications of the above geometric characterizations

163 Propositions 3.2 and 3.3 imply that the precise location of $\boldsymbol{H}_{\mathcal{M}^\perp}$ (or $\|\boldsymbol{H}_{\mathcal{M}^\perp}\|_F = \|\boldsymbol{H}\|_{\mathcal{M}^\perp}$) depends
164 on the center and the radius $r$ or $r_a$. Given a fixed $\boldsymbol{Z}_{\mathcal{M}^\perp}$, the center of the spheres remains unchanged,
165 and $r$ and $r_a$ are only affected by changes in $\boldsymbol{Z}_{\mathcal{M}}$. This observation motivates us to investigate ***how***
166 ***changes in*** $\boldsymbol{Z}_{\mathcal{M}}$ ***impact*** $\|\boldsymbol{H}\|_{\mathcal{M}^\perp}$***, i.e., the unnormalized smoothness of node features***.

167 Propositions 3.2 and 3.3 imply both ReLU and leaky ReLU reduce the distance of node features to
168 eigenspace $\mathcal{M}$, i.e. $\|\boldsymbol{H}\|_{\mathcal{M}^\perp} \leq \|\boldsymbol{Z}\|_{\mathcal{M}^\perp}$. Moreover, this inequality is independent of $\boldsymbol{Z}_{\mathcal{M}}$; consider
169 $\boldsymbol{Z}, \boldsymbol{Z}' \in \mathbb{R}^{d \times n}$ s.t. $\boldsymbol{Z}_{\mathcal{M}^\perp} = \boldsymbol{Z}'_{\mathcal{M}^\perp}$ but $\boldsymbol{Z}_{\mathcal{M}} \neq \boldsymbol{Z}'_{\mathcal{M}}$. Let $\boldsymbol{H}$ and $\boldsymbol{H}'$ be the output of $\boldsymbol{Z}$ and $\boldsymbol{Z}'$ via
170 ReLU or leaky ReLU, resp. Then we have $\|\boldsymbol{H}\|_{\mathcal{M}^\perp} \leq \|\boldsymbol{Z}\|_{\mathcal{M}^\perp}$ and $\|\boldsymbol{H}'\|_{\mathcal{M}^\perp} \leq \|\boldsymbol{Z}'\|_{\mathcal{M}^\perp}$. Since
171 $\boldsymbol{Z}_{\mathcal{M}^\perp} = \boldsymbol{Z}'_{\mathcal{M}^\perp}$, we deduce that $\|\boldsymbol{H}'\|_{\mathcal{M}^\perp} \leq \|\boldsymbol{Z}\|_{\mathcal{M}^\perp}$. In other words, when $\boldsymbol{Z}_{\mathcal{M}^\perp} = \boldsymbol{Z}'_{\mathcal{M}^\perp}$ is fixed,
172 ***changing*** $\boldsymbol{Z}_{\mathcal{M}}$ ***to*** $\boldsymbol{Z}'_{\mathcal{M}}$ ***can change the unnormalized smoothness of the output features but cannot***
173 ***change the fact that ReLU and leaky ReLU smooth node features***; we demonstrate this result in

174 Fig. 1a) in Section 4.1. Notice that without considering the nonlinear activation function, changing
175 $\boldsymbol{Z}_{\mathcal{M}}$ does not affect the unnormalized smoothness of node features measured by $\|\boldsymbol{H}\|_{\mathcal{M}^\perp}$.

176 In contrast to the unnormalized smoothness, *if one considers the normalized smoothness, we find*
177 *that adjusting $\boldsymbol{Z}_{\mathcal{M}}$ can result in a less smooth output*; we will discuss this in Section 4.1.

## 4 How Adjusting $\boldsymbol{Z}_{\mathcal{M}}$ Affects the Smoothness of the Output

179 Throughout this section, we let $\boldsymbol{Z}$ and $\boldsymbol{H}$ be the input and output of ReLU or leaky ReLU. The
180 smoothness notions based on the distance of feature to $\mathcal{M}$ or their Dirichlet energy do not account
181 for the magnitude of each dimension of the features; [4] points out that analyzing the normalized
182 smoothness of features $\boldsymbol{Z}$, given by $\|\boldsymbol{Z}\|_E/\|\boldsymbol{Z}\|_F$, is an open problem. However, these two smooth-
183 ness notions aggregate the smoothness of node features across all dimensions; when the magnitude
184 of some dimensions is much larger than others, the smoothness will be dominated by them.

185 Motivated by the discussion in Section 3.3, we study *the disparate effects of adjusting $\boldsymbol{Z}_{\mathcal{M}}$ on the*
186 *normalized and unnormalized smoothness* in this section. For the sake of simplicity, we assume
187 the graph is connected ($m = 1$); all the following results can be extended to graphs with multiple
188 connected components easily. Due to the equivalence between seminorms $\|\cdot\|_{\mathcal{M}}$ and $\|\cdot\|_E$, we
189 introduce the following definition of the dimension-wise normalized smoothness of node features.

**Definition 4.1.** Let $\boldsymbol{Z} \in \mathbb{R}^{d \times n}$ be the features over $n$ nodes with $\boldsymbol{z}^{(i)} \in \mathbb{R}^n$ being its $i^{th}$ row, i.e., the $i^{th}$ dimension of the features over all nodes. We define the normalized smoothness of $\boldsymbol{z}^{(i)}$ as follows:

$$s(\boldsymbol{z}^{(i)}) := \|\boldsymbol{z}^{(i)}_{\mathcal{M}}\|/\|\boldsymbol{z}^{(i)}\|,$$

190 where we set $s(\boldsymbol{z}^{(i)}) = 1$ when $\boldsymbol{z}^{(i)} = \boldsymbol{0}$.

191 *Remark* 4.2. Notice that the normalized smoothness $s(\boldsymbol{z}^{(i)}) = \|\boldsymbol{z}^{(i)}_{\mathcal{M}}\|/\|\boldsymbol{z}^{(i)}\|$ is closely related to the
192 ratio between the smooth and non-smooth components of node features $\|\boldsymbol{z}^{(i)}_{\mathcal{M}}\|/\|\boldsymbol{z}^{(i)}_{\mathcal{M}^\perp}\|$.

193 The graph is connected implies that $\boldsymbol{z}^{(i)}_{\mathcal{M}} = \langle \boldsymbol{z}^{(i)}, \boldsymbol{e}_1\rangle \boldsymbol{e}_1$ and $\|\boldsymbol{z}^{(i)}_{\mathcal{M}}\| = |\langle \boldsymbol{z}^{(i)}, \boldsymbol{e}_1\rangle|$. Without ambiguity,
194 we write $\boldsymbol{z}$ for $\boldsymbol{z}^{(i)}$ and $\boldsymbol{e}$ for $\boldsymbol{e}_1$ – the eigenvector of $\boldsymbol{G}$ associated with the eigenvalue 1. Moreover,
195 we have

$$s(\boldsymbol{z}) = \frac{\|\boldsymbol{z}_{\mathcal{M}}\|}{\|\boldsymbol{z}\|} = \frac{|\langle \boldsymbol{z}, \boldsymbol{e}\rangle|}{\|\boldsymbol{z}\|} = \frac{|\langle \boldsymbol{z}, \boldsymbol{e}\rangle|}{\|\boldsymbol{z}\| \cdot \|\boldsymbol{e}\|} \Rightarrow 0 \leq s(\boldsymbol{z}) \leq 1, \tag{4}$$

It is evident that *the larger $s(\boldsymbol{z})$ is, the smoother the node feature $\boldsymbol{z}$ is*[1]. In fact, we have

$$s(\boldsymbol{z})^2 + \left(\frac{\|\boldsymbol{z}\|_{\mathcal{M}^\perp}}{\|\boldsymbol{z}\|}\right)^2 = \frac{\|\boldsymbol{z}_{\mathcal{M}}\|^2}{\|\boldsymbol{z}\|^2} + \frac{\|\boldsymbol{z}_{\mathcal{M}^\perp}\|^2}{\|\boldsymbol{z}\|^2} = 1,$$

196 where $\|\boldsymbol{z}\|_{\mathcal{M}^\perp}/\|\boldsymbol{z}\|$ decreases as $s(\boldsymbol{z})$ increases.

To discuss how the smoothness $s(\boldsymbol{h}) = s(\sigma(\boldsymbol{z}))$ or $s(\sigma_a(\boldsymbol{z}))$ can be adjusted by changing $\boldsymbol{z}_{\mathcal{M}}$, we consider the function

$$\boldsymbol{z}(\alpha) = \boldsymbol{z} - \alpha\boldsymbol{e}.$$

It is clear that

$$\boldsymbol{z}(\alpha)_{\mathcal{M}^\perp} = \boldsymbol{z}_{\mathcal{M}^\perp} \text{ and } \boldsymbol{z}(\alpha)_{\mathcal{M}} = \boldsymbol{z}_{\mathcal{M}} - \alpha\boldsymbol{e},$$

197 where we see that $\alpha$ only alters $\boldsymbol{z}_{\mathcal{M}}$ while pre-
198 serves $\boldsymbol{z}_{\mathcal{M}^\perp}$. Moreover, it is evident that

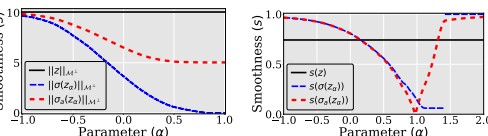

a) Smoothness      b) Normalized smoothness

Figure 1: Contrasting the effects of varying parameter $\alpha$ on the smoothness and normalized smoothness of output features $\sigma(\boldsymbol{z}_\alpha)$ and $\sigma_a(\boldsymbol{z}_\alpha)$. The discontinuity of $s(\sigma(\boldsymbol{z}_\alpha))$ in b) comes from the definition of normalized smoothness. Note that $s(\boldsymbol{z}) = 1$ if $\boldsymbol{z} = \boldsymbol{0}$, and $\sigma(\boldsymbol{z}_\alpha)$ can become $\boldsymbol{0}$ when $\alpha$ is large enough.

$$s(\boldsymbol{z}(\alpha)) = \sqrt{1 - \frac{\|\boldsymbol{z}(\alpha)_{\mathcal{M}^\perp}\|^2}{\|\boldsymbol{z}(\alpha)\|^2}} = \sqrt{1 - \frac{\|\boldsymbol{z}_{\mathcal{M}^\perp}\|^2}{\|\boldsymbol{z}(\alpha)\|^2}}.$$

199 It follows that $s(\boldsymbol{z}(\alpha)) = 1$ if and only if $\boldsymbol{z}_{\mathcal{M}^\perp} = \boldsymbol{0}$ (include the case $\boldsymbol{z} = \boldsymbol{0}$), showing that when
200 $\boldsymbol{z}_{\mathcal{M}^\perp} = \boldsymbol{0}$, the vector $\boldsymbol{z}$ is the smoothest one.

### 4.1 The disparate effects of $\alpha$ on $\|\cdot\|_{\mathcal{M}^\perp}$ and $s(\cdot)$: Empirical results

202 Let us empirically study possible values that the unnormalized smoothness $\|\sigma(\boldsymbol{z}(\alpha))\|_{\mathcal{M}^\perp}$,
203 $\|\sigma_a(\boldsymbol{z}(\alpha))\|_{\mathcal{M}^\perp}$ and the normalized smoothness $s(\sigma(\boldsymbol{z}(\alpha)))$, $s(\sigma_a(\boldsymbol{z}(\alpha)))$ can take when $\alpha$ varies.

---

[1]Here, $\boldsymbol{z} \in \mathbb{R}^n$ is a vector whose $i^{th}$ entry is the 1D feature associated with node $i$.

We denote $\boldsymbol{z}_\alpha := \boldsymbol{z}(\alpha) = \boldsymbol{z} - \alpha \boldsymbol{e}$. We consider a connected synthetic graph with 100 nodes, and each node is assigned a random degree between 2 to 10. Then we assign an initial node feature $\boldsymbol{z} \in \mathbb{R}^{100}$, sampled uniformly on the interval $[-1.5, 1.5]$, to the graph with each node feature being a scalar. Also, we compute $\boldsymbol{e}$ by the formula $\boldsymbol{e} = \tilde{\boldsymbol{D}}^{\frac{1}{2}} \boldsymbol{u} / \|\tilde{\boldsymbol{D}}^{\frac{1}{2}} \boldsymbol{u}\|$ from Proposition 2.1, where $\boldsymbol{u} \in \mathbb{R}^{100}$ is the vector whose entries are all ones and $\tilde{\boldsymbol{D}}$ is the (augmented) degree matrix. We examine two different smoothness notions for the input $\boldsymbol{z}$ and the output $\sigma(\boldsymbol{z}_\alpha)$ and $\sigma_a(\boldsymbol{z}_\alpha)$, where the smoothness is measured for various values of the smoothness control parameter $\alpha \in [-1.5, 1.5]$. In Fig. 1a), we study the unnormalized smoothness measured by $\|\cdot\|_{\mathcal{M}^\perp}$; we see that $\|\sigma(\boldsymbol{z}_\alpha)\|_{\mathcal{M}^\perp}$ and $\|\sigma_a(\boldsymbol{z}_\alpha)\|_{\mathcal{M}^\perp}$ are always no greater than $\|\boldsymbol{z}\|_{\mathcal{M}^\perp}$. This coincides with the discussion in Section 3.3; adjusting the projection of $\boldsymbol{z}$ onto the eigenspace $\mathcal{M}$ can not change the fact that $\|\sigma(\boldsymbol{z}_\alpha)\|_{\mathcal{M}^\perp} \leq \|\boldsymbol{z}\|_{\mathcal{M}^\perp}$ and $\|\sigma_a(\boldsymbol{z}_\alpha)\|_{\mathcal{M}^\perp} \leq \|\boldsymbol{z}\|_{\mathcal{M}^\perp}$. Nevertheless, an interesting result is that ***altering the eigenspace projection can adjust the unnormalized smoothness of the output***: notice that altering the eigenspace projection does not change its distance to $\mathcal{M}$, i.e., the smoothness of the input is unchanged, but the smoothness of the output after activation function can be changed.

In contrast, when studying the normalized smoothness $s(\cdot)$ in Fig. 1b), we find that $s(\sigma(\boldsymbol{z}(\alpha)))$ and $s(\sigma_a(\boldsymbol{z}(\alpha)))$ can be adjusted by $\alpha$ to values smaller than $s(\boldsymbol{z})$. More precisely, we see that by adjusting $\alpha$, $s(\sigma(\boldsymbol{z}(\alpha)))$ and $s(\sigma_a(\boldsymbol{z}(\alpha)))$ can achieve most of the values in $[0, 1]$. In other words, both smoother and less smooth features can be obtained by adjusting $\alpha$.

## 4.2 Theoretical results on the smooth effects of ReLU and leaky ReLU

In this subsection, we build theoretical understandings of the above empirical findings on the achievable smoothness shown in Fig. 1. Notice that if $\boldsymbol{z}_{\mathcal{M}^\perp} = \boldsymbol{0}$, the inequalities presented in Propositions 3.2 and 3.3 indicate that $\|\sigma(\boldsymbol{z}(\alpha))\|_{\mathcal{M}^\perp}$ and $\|\sigma_a(\boldsymbol{z}(\alpha))\|_{\mathcal{M}^\perp}$ vanish. So we have $s(\sigma(\boldsymbol{z}(\alpha))) = 1$ for any $\alpha$ when $\boldsymbol{z}_{\mathcal{M}^\perp} = \boldsymbol{0}$. Then we may assume $\boldsymbol{z}_{\mathcal{M}^\perp} \neq \boldsymbol{0}$ for the following study.

**Proposition 4.3** (ReLU). *Suppose $\boldsymbol{z}_{\mathcal{M}^\perp} \neq \boldsymbol{0}$. Let $\boldsymbol{h}(\alpha) = \sigma(\boldsymbol{z}(\alpha))$ with $\sigma$ being ReLU, then*

$$\min_\alpha s(\boldsymbol{h}(\alpha)) = \sqrt{\frac{\sum_{x_i = \max \boldsymbol{x}} d_i}{\sum_{j=1}^n d_j}} \ \text{ and } \ \max_\alpha s(\boldsymbol{h}(\alpha)) = 1,$$

*where $\boldsymbol{x} := \tilde{\boldsymbol{D}}^{-\frac{1}{2}} \boldsymbol{z}$, $\max \boldsymbol{x} = \max_{1 \leq i \leq n} x_i$, and $\tilde{\boldsymbol{D}}$ is the augmented degree matrix with diagonals $d_1, d_2, \ldots, d_n$. In particular, the normalized smoothness $s(\boldsymbol{h}(\alpha))$ is monotone increasing as $\alpha$ decreases whenever $\alpha < \|\tilde{\boldsymbol{D}}^{\frac{1}{2}} \boldsymbol{u}_n\| \max \boldsymbol{x}$ and it has range $[\min_\alpha s(\boldsymbol{h}(\alpha)), 1]$.*

**Proposition 4.4** (Leaky ReLU). *Suppose $\boldsymbol{z}_{\mathcal{M}^\perp} \neq \boldsymbol{0}$. Let $\boldsymbol{h}(\alpha) = \sigma_a(\boldsymbol{z}(\alpha))$ with $\sigma_a$ being leaky ReLU, then (1) $\min_\alpha s(\boldsymbol{h}(\alpha)) = 0$, and (2) $\sup_\alpha s(\boldsymbol{h}(\alpha)) = 1$ and $s(\boldsymbol{h}(\alpha))$ has range $[0, 1)$.*

Proposition 4.4 also holds for other variants of ReLU, e.g., ELU[2] and SELU[3].; see Appendix C. We summarize Propositions 3.2, 3.3, 4.3, and 4.4 in the following corollary, which qualitatively explains the empirical results in Fig. 1.

**Corollary 4.5.** *Suppose $\boldsymbol{z}_{\mathcal{M}^\perp} \neq \boldsymbol{0}$. Let $\boldsymbol{h}(\alpha) = \sigma(\boldsymbol{z}(\alpha))$ or $\sigma_a(\boldsymbol{z}(\alpha))$ with $\sigma$ being ReLU and $\sigma_a$ being leaky ReLU. Then we have $\|\boldsymbol{z}\|_{\mathcal{M}^\perp} \geq \|\boldsymbol{h}(\alpha)\|_{\mathcal{M}^\perp}$ for any $\alpha \in \mathbb{R}$; however, $s(\boldsymbol{h}(\alpha))$ can be smaller than, larger than, or equal to $s(\boldsymbol{z})$ for different values of $\alpha$.*

Propositions 4.3 and 4.4, and Corollary 4.5, provide a theoretical basis for the empirical results in Fig. 1. Moreover, our results indicate that for any given vector $\boldsymbol{z}$, altering $\boldsymbol{z}_{\mathcal{M}}$ can change both the unnormalized and the normalized smoothness of the output vector $\boldsymbol{h} = \sigma(\boldsymbol{z})$ or $\sigma_a(\boldsymbol{z})$. In particular, the normalized smoothness of $\boldsymbol{h} = \sigma(\boldsymbol{z})$ or $\sigma_a(\boldsymbol{z})$ can be adjusted to any value in the range shown in Propositions 4.3 and 4.4. This provides us with insights to control the smoothness of features to improve the performance of GCN and we will discuss this in the next section.

## 5 Controlling Smoothness of Node Features

We do not know how smooth features are ideal for a given node classification task. Nevertheless, our theory indicates that both normalized and unnormalized smoothness of the output of each GCL can be adjusted by altering the input's projection onto $\mathcal{M}$. As such, we propose the following learnable smoothness control term to modulate the smoothness of each dimension of the learned node features

$$\boldsymbol{B}_{\boldsymbol{\alpha}}^l = \sum_{i=1}^m \boldsymbol{\alpha}_i^l \boldsymbol{e}_i^\top, \tag{5}$$

---

[2]The ELU function is defined by $f(x) = \max(x, 0) + \min(0, a \cdot (e^x - 1))$ where $a > 0$.

[3]The SELU function is defined by $f(x) = c(\max(x, 0) + \min(0, a \cdot (e^x - 1)))$ where $a, c > 0$.

where $l$ is the layer index, $\{e_i\}_{i=1}^m$ is the orthonormal basis of the eigenspace $\mathcal{M}$, and $\boldsymbol{\alpha}^l := \{\boldsymbol{\alpha}_i^l\}_{i=1}^m$ is a collection of learnable vectors with $\boldsymbol{\alpha}_i^l \in \mathbb{R}^d$ being approximated by a multi-layer perceptron (MLP). The detailed configuration of $\boldsymbol{\alpha}_i^l$ will be specified in each experiment later. One can see that $\boldsymbol{B}_{\boldsymbol{\alpha}}^l$ always lies in $\mathbb{R}^d \otimes \mathcal{M}$. We integrate SCT into GCL, resulting in

$$\boldsymbol{H}^l = \sigma(\boldsymbol{W}^l \boldsymbol{H}^{l-1} \boldsymbol{G} + \boldsymbol{B}_{\boldsymbol{\alpha}}^l). \tag{6}$$

We call the corresponding model GCN-SCT. Again, the idea is that ***we alter the component in eigenspace to control the smoothness of features***. Each dimension of $\boldsymbol{H}^l$ can be smoother, less smooth, or the same as $\boldsymbol{H}^{l-1}$ in normalized smoothness, though $\boldsymbol{H}^l$ gets closer to $\mathcal{M}$ than $\boldsymbol{H}^{l-1}$.

To design SCT, we introduce a learnable matrix $\boldsymbol{A}^l \in \mathbb{R}^{d \times m}$ for layer $l$, whose columns are $\boldsymbol{\alpha}_i^l$, where $m$ is the dimension of the eigenspace $\mathcal{M}$ and $d$ is the dimension of the features. We observe in our experiments that the SCT performs best when informed by degree pooling over the subcomponents of the graph. The matrix of the orthogonal basis vectors, denoted by $\boldsymbol{Q} := [\boldsymbol{e}_1, \ldots, \boldsymbol{e}_m] \in \mathbb{R}^{n \times m}$, is used to perform pooling $\boldsymbol{H}^l \boldsymbol{Q}$ for input $\boldsymbol{H}^l$. In particular, we let $\boldsymbol{A}^l = \boldsymbol{W} \odot (\boldsymbol{H}^l \boldsymbol{Q})$, where $\boldsymbol{W} \in \mathbb{R}^{d \times m}$ is learnable and performs pooling over $\boldsymbol{H}^l$ using the eigenvectors $\boldsymbol{Q}$. The second architecture uses a residual connection with hyperparameter $\beta_l = \log(\theta/l + 1)$ and learnable matrices $\boldsymbol{W}_0, \boldsymbol{W}_1 \in \mathbb{R}^{d \times d}$ and the softmax function $\phi$. Resulting in $\boldsymbol{A}^l = \phi(\boldsymbol{H}^l \boldsymbol{Q}) \odot (\beta_l \boldsymbol{W}_0 \boldsymbol{H}^0 \boldsymbol{Q} + (1 - \beta_l) \boldsymbol{W}_1 \boldsymbol{H}^l \boldsymbol{Q})$. In Section 6, we use the first architecture for GCN-SCT as GCN uses only $\boldsymbol{H}^l$ information at each layer. We use the second architecture for GCNII-SCT and EGNN-SCT which use both $\boldsymbol{H}^0$ and $\boldsymbol{H}^l$ information at each layer. There are two particular advantages of the above design of SCT: (1) it can effectively change the normalized smoothness of the learned features, and (2) it is computationally efficient since we only use the eigenvectors corresponding to the eigenvalue 1 of matrix $\boldsymbol{G}$, which is determined based on the connectivity of the graph.

## 5.1 Integrating SCT into other GCN-style models

In this subsection, we present other usages of the proposed SCT. Due to the page limit, we carefully select two other most representative models. The first example is GCNII [6], GCNII extends GCN to express an arbitrary polynomial filter rather than the Laplacian polynomial filter and achieves state-of-the-art (SOTA) performance among GCN-style models on various tasks [6, 23], and we aim to show that SCT can even improve the accuracy of the GCN-style model that achieves SOTA performance on many node classification tasks. The second example is energetic GNN (EGNN) [40], which controls the smoothness of node features by constraining the lower and upper bounds of the Dirichlet energy of features and assuming the activation function is linear. In this case, we aim to show that our new theoretical understanding of the role of activation functions and the proposed SCT can boost the performance of EGNN with considering nonlinear activation functions.

**GCNII.** Each GCNII layer uses a skip connection to the initial layer $\boldsymbol{H}^0$ and given as follows:

$$\boldsymbol{H}^l = \sigma\big(((1 - \alpha_l)\boldsymbol{H}^{l-1}\boldsymbol{G} + \alpha_l \boldsymbol{H}^0)((1 - \beta_l)\boldsymbol{I} + \beta_l \boldsymbol{W}^l)\big),$$

where $\alpha_l, \beta_l \in (0, 1)$ are learnable scalars. We integrate SCT $\boldsymbol{B}_{\boldsymbol{\alpha}}^l$ into GCNII, resulting in the following GCNII-SCT layers

$$\boldsymbol{H}^l = \sigma\big(((1 - \alpha_l)\boldsymbol{H}^{l-1}\boldsymbol{G} + \alpha_l \boldsymbol{H}^0)((1 - \beta_l)\boldsymbol{I} + \beta_l \boldsymbol{W}^l) + \boldsymbol{B}_{\boldsymbol{\alpha}}^l\big),$$

where the residual connection and identity mapping are consistent with GCNII.

**EGNN.** Each EGNN layer can be written as follows:

$$\boldsymbol{H}^l = \sigma\big(\boldsymbol{W}^l(c_1 \boldsymbol{H}^0 + c_2 \boldsymbol{H}^{l-1} + (1 - c_{\min})\boldsymbol{H}^{l-1}\boldsymbol{G})\big), \tag{7}$$

where $c_1, c_2$ are learnable weights that satisfy $c_1 + c_2 = c_{\min}$ with $c_{\min}$ being a hyperparameter. To constrain Dirichlet energy, EGNN initializes trainable weights $\boldsymbol{W}^l$ as a diagonal matrix with explicit singular values and regularizes them to keep the orthogonality during the model training. Ignoring the activation function $\sigma$, $\boldsymbol{H}^l$ – node features at layer $l$ of EGNN satisfies

$$c_{\min}\|\boldsymbol{H}^0\|_E \leq \|\boldsymbol{H}^l\|_E \leq c_{\max}\|\boldsymbol{H}^0\|_E,$$

where $c_{\max}$ is the square of the maximal singular value of the initialization of $\boldsymbol{W}^1$. Similarly, we modify EGNN to result in the following EGNN-SCT layer

$$\boldsymbol{H}^l = \sigma\big(\boldsymbol{W}^l((1 - c_{\min})\boldsymbol{H}^{l-1}\boldsymbol{G} + c_1 \boldsymbol{H}^0 + c_2 \boldsymbol{H}^{l-1}) + \boldsymbol{B}_{\boldsymbol{\alpha}}^l\big),$$

where everything remains the same as the EGNN layer except that we add our proposed SCT $\boldsymbol{B}_{\boldsymbol{\alpha}}^l$.

## 6 Experiments

In this section, we comprehensively demonstrate the effects of SCT – in the three most representative GCN-style models discussed in Section 5 – using various node classification benchmarks. The purpose of all experiments in this section is to verify the efficacy of the proposed SCT – motivated by our theoretical results – for GCN-style models. We consider the citation datasets (Cora, Citeseer, PubMed, Coauthor-Physics, Ogbn-arxiv), web knowledge-base datasets (Cornell, Texas, Wisconsin), and Wikipedia network datasets (Chameleon, Squirrel). We provide additional dataset details in Appendix D.1. We implement baseline GCN [20] and GCNII [6] (without weight sharing) using PyG (Pytorch Geometric) [10]. Baseline EGNN [40] is implemented using the public code[4].

### 6.1 Node feature trajectory

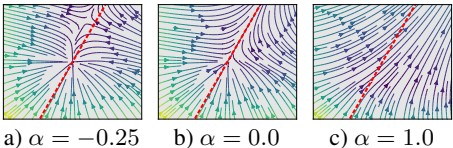

We visualize the trajectory of the node features, following [27], for a graph with two nodes connected by an edge and 1D node feature. In this case, (6) becomes $\boldsymbol{h}^1 = \sigma(w\boldsymbol{h}^0\boldsymbol{G} + \boldsymbol{b}_\alpha)$, where $w = 1.2$ in our experiment, $\boldsymbol{h}^0, \boldsymbol{h}^1, \boldsymbol{b}_\alpha \in \mathbb{R}^2$, and $\boldsymbol{G} \in \mathbb{R}^{2\times2}$. We use a matrix $\boldsymbol{G} = [0.592, 0.194; 0.194, 0.908]$ whose largest eigenvalue is 1. Twenty initial node feature vectors $\boldsymbol{h}^0$ are sampled evenly in the domain $[-1, 1] \times [-1, 1]$. Fig. 2 shows the trajectories in relation to the eigenspace $\mathcal{M}$ (red dashed line). In Fig 2a), one can see that some trajectories do not directly converge to $\mathcal{M}$. In Fig. 2b) when $\alpha = 0.0$, GCL is recovered and all trajectories converge to $\mathcal{M}$. In Fig. 2c), large values of $\alpha$ enable the features to significantly deviate from $\mathcal{M}$ initially. We observe that the parameter $\alpha$ can effectively change the trajectory of features.

a) $\alpha = -0.25$    b) $\alpha = 0.0$    c) $\alpha = 1.0$

Figure 2: Node feature trajectories, with colorized magnitude, for varying smoothness control parameter $\alpha$. For classical GCN b), the node features converge to the eigenspace $\mathcal{M}$ (red dashed line).

| Layers | 2 | 4 | 16 | 32 |
|---|---|---|---|---|
| **Cora** | | | | |
| GCN/GCN-SCT | 81.1/**82.9** | 80.4/**82.8** | 64.9/**71.4** | 60.3/**67.2** |
| GCNII/GCNII-SCT | 82.2/**83.8** | 82.6/**84.3** | 84.6/**84.8** | 85.4/**85.5** |
| EGNN/EGNN-SCT | 83.2/**84.1** | 84.2/**84.5** | **85.4**/83.3 | **85.3**/82.0 |
| **Citeseer** | | | | |
| GCN/GCN-SCT | **70.3**/69.9 | 67.6/**67.7** | 18.3/**55.4** | 25.0/**51.0** |
| GCNII/GCNII-SCT | 68.2/**72.8** | 68.9/**72.8** | 72.9/**73.8** | **73.4**/**73.4** |
| EGNN/EGNN-SCT | 72.0/**73.1** | 71.9/**72.0** | 72.4/**72.6** | 72.3/**72.9** |
| **PubMed** | | | | |
| GCN/GCN-SCT | 79.0/**79.8** | 76.5/**78.4** | 40.9/**76.1** | 22.4/**77.0** |
| GCNII/GCNII-SCT | 78.2/**79.7** | 78.8/**80.1** | 80.2/**80.7** | 79.8/**80.7** |
| EGNN/EGNN-SCT | 79.2/**79.8** | 79.5/**80.4** | 80.1/**80.3** | 80.0/**80.4** |
| **Coauthor-Physics** | | | | |
| GCN/GCN-SCT | 92.4/**92.6** $\pm$ **1.6** | 92.1/**92.5** $\pm$ **5.9** | 13.5/**50.9** $\pm$ **15.0** | 13.1/**43.6** $\pm$ **16.0** |
| GCNII/GCNII-SCT | 92.5/**94.4** $\pm$ **0.4** | 92.9/**94.2** $\pm$ **0.3** | 92.9/**93.7** $\pm$ **0.7** | 92.9/**94.1** $\pm$ **0.3** |
| EGNN/EGNN-SCT | 92.6/**93.9** $\pm$ **0.7** | 92.9/**94.1** $\pm$ **0.4** | 93.1/**94.0** $\pm$ **0.7** | 93.3/**93.8** $\pm$ **1.3** |
| **Ogbn-arxiv** | | | | |
| GCN/GCN-SCT | 70.4/**72.1** $\pm$ **0.3** | 71.7/**72.7** $\pm$ **0.3** | 70.6/**72.3** $\pm$ **0.2** | 68.5/**72.3** $\pm$ **0.3** |
| GCNII/GCNII-SCT | 70.1/**72.0** $\pm$ **0.3** | 71.4/**72.2** $\pm$ **0.2** | 71.5/**72.4** $\pm$ **0.3** | 70.5/**72.1** $\pm$ **0.3** |
| EGNN/EGNN-SCT | 68.4/**68.5** $\pm$ **0.6** | 71.1/**71.3** $\pm$ **0.5** | 72.7/**72.8** $\pm$ **0.5** | **72.7**/72.3 $\pm$ 0.5 |

Table 1: Accuracy for models of varying depth. We note vanishing gradients occur but not over-smoothing for the accuracy drop using GCN-SCT with 16 or 32 layers. For Cora, Citeseer, and PubMed, we use a fixed split with a single forward pass following [6]; only test accuracy is available in these experiments. For Coauthor-Physics and Ogbn-arxiv, we use the splits from [40]; both test accuracy and standard deviation are reported. The baseline results are copied from [6, 40] where the standard deviation was not reported. (Unit:%)

### 6.2 Baseline comparisons for node classification

**Citation networks.** We compare the three representative models discussed in Section 5, of different depths, with and without SCT in Table 1. This task uses the citation datasets with fixed splits from [35] for Cora, Citeseer, and Pubmed and splits from [40] for Coauthor-Physics and Ogbn-arxiv; a detailed description of these datasets and splits are provided in Appendix D. Following [6], we use a single training pass to minimize the negative log-likelihood loss using the Adam optimizer [19], with 1500 maximum epochs, and 100 epochs of patience. A grid search for possible hyperparameters is listed in Table 5 in Appendix D. We accelerate the hyperparameter search by applying a Bayesian meta-learning algorithm [2] which minimizes the validation loss, and we run the search for 200 iterations per model. In particular, Table 1 presents the best test accuracy between ReLU and leaky ReLU for GCN, GCNII, and all three models with SCT[5]. For the baseline EGNN, we follow [40] using SReLU, a particular activation used for EGNN in [40]. These results show that SCT can boost

---

[4]https://github.com/Kaixiong-Zhou/EGNN

[5]A comparison of the results using ReLU and leaky ReLU is presented in Appendix D.

the classification accuracy of baseline models; in particular, the improvement can be remarkable for GCN and GCNII. However, EGNN-SCT (using ReLU or leaky ReLU) performs occasionally worse than EGNN (using SReLU), and this is because of the choice of activation functions. In Appendix D.3, we report the results of EGNN-SCT using SReLU, showing that EGNN-SCT outperforms EGNN in all tasks. In fact, SReLU is a shifted version of ReLU, and our theory for ReLU applies to SReLU as well. The model size and computational time are reported in Table 4 in the appendix.

Table 1 also shows that even with SCT, the accuracy of GCN drops when the depth is 16 or 32. This motivates us to investigate the smoothness of the node features learned by GCN and GCN-SCT. Fig. 3 plots the heatmap of the normalized smoothness of each dimension of the learned node features learned by GCN and GCN-SCT with 32 layers for Citeseer node classification. In these plots, the horizontal and vertical dimensions denote the feature dimension and the layer of the model, resp. We notice that the normalized smoothness of each dimension of the features – from layers 14 to 32 learned by GCN – closes to 1, confirming that deep GCN learns homogeneous features. In contrast, the features learned by GCN-SCT are inhomogeneous, as shown in Fig. 3b). Therefore, we believe the performance degradation of deep GCN-SCT is due to other factors. Compared to GCNII/GCNII-SCT and EGNN/EGNN-SCT, GCN-SCT does not use skip connections, which is known to help avoid vanishing gradients in training deep neural networks [16, 17]. In Appendix D.3, we show that training GCN and GCN-SCT do suffer from the vanishing gradient issue; however, the other models do not. Besides Citeseer, we notice similar behavior occurs for training GCN and GCN-SCT for Cora and Coauthor-Physics node classification tasks.

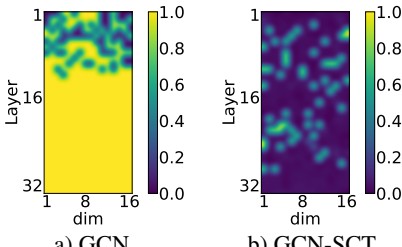

a) GCN  b) GCN-SCT

**Other datasets.** We further compare different models trained on different datasets using 10-fold cross-validation and fixed $48/32/20\%$ splits following [28]. Table 2 compares GCN and GCNII with and without SCT, using leaky ReLU, for classifying five heterophilic node classification datasets. We exclude EGNN as these heterophilic datasets are not considered in [40]. We report the average accuracy of GCN and GCNII from [6]. We tune all other models using a Bayesian meta-learning algorithm to maximize the mean validation accuracy. We report the best test accuracy for each model of depth searched over the set $\{2, 4, 8, 16, 32\}$. SCT can significantly improve the classification accuracy of the baseline models. Table 2 also contrasts the computational time (on Tesla T4 GPUs from Google Colab) per epoch of models that achieve the best test accuracy; the models using SCT can even save computational time to achieve the best accuracy which is because

Figure 3: The normalized smoothness – of each dimension of the feature vectors at a given layer – for a) GCN and b) GCN-SCT on the Citeseer dataset with 32 layers and 16 hidden dimensions. GCN features become entirely smooth since layer 14, while GCN-SCT controls the smoothness for each feature at any depth. Horizontal and vertical axes represent the index of the feature dimension and the intermediate layer, resp.

the best accuracy is achieved at a moderate depth (Table 8 in Appendix D.4 lists the mean and standard deviation for the test accuracies on all five datasets. Table 9 in Appendix D.4 lists the computational time per epoch for each model of depth 8, showing that using SCT only takes a small amount of computational overhead.

| Cornell | Texas | Wisconsin | Chameleon | Squirrel |
|---|---|---|---|---|
| 52.70/**55.95** (0.7/1.8) | 52.16/**62.16** (0.7/0.8) | 45.88/**54.71** (0.7/0.8) | 28.18/**38.44** (0.6/0.7) | 23.96/**35.31** (1.6/4.0) |
| 74.86/**75.41** (2.0/2.0) | 69.46/**83.34** (3.1/2.0) | 74.12/**86.08** (2.0/1.5) | 60.61/**64.52** (1.5/1.3) | 38.47/**47.51** (5.5/3.7) |

Table 2: Mean test accuracy and average computational time per epoch (in the parenthesis) for the WebKB and WikipediaNetwork datasets with fixed $48/32/20\%$ splits. First row: GCN/GCN-SCT. Second row: GCNII/GCNII-SCT. (Unit:% for accuracy and $\times 10^{-2}$ second for computational time.)

# 7 Concluding Remarks

In this paper, we establish a geometric characterization of how ReLU and leaky ReLU affect the smoothness of the GCN features. We further study the dimension-wise normalized smoothness of the learned node features, showing that activation functions not only smooth node features but also can reduce or preserve the normalized smoothness of the features. Our theoretical findings inform the design of a simple yet effective SCT for GCN. The proposed SCT can change the smoothness, in terms of both normalized and unnormalized smoothness, of the learned node features by GCN.

**Limitations:** Our proposed SCT provides provable guarantees for controlling the smoothness of features learned by GCN and related models. A key aspect to establish our theoretical results is demonstrating that, without SCT, the features of the vanilla model tend to be overly smooth; without this condition, SCT cannot ensure performance guarantees.

# 8 Broader Impacts

Our paper focuses on developing new theoretical understandings of the smoothness of node features learned by graph convolutional networks. The paper is mainly theoretical. We do not see any potential ethical issues in our research; all experiments are carried out using existing benchmark settings and datasets.

Our paper brings new insights into building new graph neural networks with improved performance over existing models, which is crucial for many applications. In particular, for applications where graph neural network is the method of choice. We expect our approach to play a role in material science and biophysics applications.

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

# Appendix for "Learning to Control the Smoothness of GCN Features"

## A  Details of Notations

For two vectors $\boldsymbol{u} = (u_1, u_2, \ldots, u_d)$ and $\boldsymbol{v} = (v_1, v_2, \ldots, v_d)$, their inner product is defined as

$$\langle \boldsymbol{u}, \boldsymbol{v} \rangle = \sum_{i=1}^{d} u_i v_i,$$

their Hadamard product is defined as

$$\boldsymbol{u} \odot \boldsymbol{v} = (u_1 v_1, u_2 v_2, \ldots, u_d v_d),$$

and their Kronecker product is defined as

$$\boldsymbol{u} \otimes \boldsymbol{v} = \boldsymbol{u} \boldsymbol{v}^\top = \begin{pmatrix} u_1 v_1 & u_1 v_2 & \ldots & u_1 v_d \\ u_2 v_1 & u_2 v_2 & \ldots & u_2 v_d \\ \vdots & \vdots & \ddots & \vdots \\ u_d v_1 & u_d v_2 & \ldots & u_d v_d \end{pmatrix}.$$

 The Kronecker product can be defined for two vectors of different lengths in a similar manner as above.

## B  Proofs in Section 3

 First, we prove that the two smoothness notions used in [27, 4] are two equivalent seminorms, i.e., we prove Proposition 3.1 below.

*Proof of Proposition 3.1.* The matrix $\boldsymbol{H}$ can be decomposed as $\boldsymbol{H} = \sum_{i=1}^{n} \boldsymbol{H} \boldsymbol{e}_i \boldsymbol{e}_i^\top$, where each $\boldsymbol{e}_i$ is the eigenvector of $\boldsymbol{G}$ associated with eigenvalue $\lambda_i$. This indicates that

$$\begin{aligned}
\boldsymbol{H} \tilde{\Delta} &= \boldsymbol{H}(\boldsymbol{I} - \boldsymbol{G}) \\
&= \sum_{i=1}^{n} \boldsymbol{H} \boldsymbol{e}_i \boldsymbol{e}_i^\top (\boldsymbol{I} - \boldsymbol{G}) \\
&= \sum_{i=1}^{n} (\boldsymbol{H} \boldsymbol{e}_i \boldsymbol{e}_i^\top - \boldsymbol{H} \boldsymbol{e}_i \boldsymbol{e}_i^\top \boldsymbol{G}) \\
&= \sum_{i=1}^{n} (\boldsymbol{H} \boldsymbol{e}_i \boldsymbol{e}_i^\top - \boldsymbol{H} \boldsymbol{e}_i (\lambda_i \boldsymbol{e}_i)^\top) \\
&= \sum_{i=1}^{n} (1 - \lambda_i) \boldsymbol{H} \boldsymbol{e}_i \boldsymbol{e}_i^\top \\
&= \sum_{i=m+1}^{n} (1 - \lambda_i) \boldsymbol{H} \boldsymbol{e}_i \boldsymbol{e}_i^\top.
\end{aligned}$$

Then using the fact that $1 - \lambda_i \geq 0$ for each $i$, we obtain

$$
\begin{aligned}
\|\boldsymbol{H}\|_E^2 &= \mathrm{Trace}(\boldsymbol{H}\tilde{\boldsymbol{\Delta}}\boldsymbol{H}^\top) \\
&= \mathrm{Trace}\Big( \sum_{i=m+1}^{n} (1-\lambda_i)\boldsymbol{H}\boldsymbol{e}_i\boldsymbol{e}_i^\top (\sum_{j=1}^{n} \boldsymbol{H}\boldsymbol{e}_j\boldsymbol{e}_j^\top)^\top \Big) \\
&= \mathrm{Trace}\Big( \sum_{i=m+1}^{n}\sum_{j=1}^{n} (1-\lambda_i)\boldsymbol{H}\boldsymbol{e}_i\boldsymbol{e}_i^\top\boldsymbol{e}_j\boldsymbol{e}_j^\top\boldsymbol{H}^\top \Big) \\
&= \mathrm{Trace}\Big( \sum_{i=m+1}^{n} (1-\lambda_i)\boldsymbol{H}\boldsymbol{e}_i\boldsymbol{e}_i^\top\boldsymbol{e}_i\boldsymbol{e}_i^\top\boldsymbol{H}^\top \Big) \\
&= \mathrm{Trace}\Big( \sum_{i=m+1}^{n} \sqrt{1-\lambda_i}\boldsymbol{H}\boldsymbol{e}_i\boldsymbol{e}_i^\top\boldsymbol{e}_i\boldsymbol{e}_i^\top\boldsymbol{H}^\top\sqrt{1-\lambda_i} \Big) \\
&= \mathrm{Trace}\Big( \sum_{i=m+1}^{n} \sqrt{1-\lambda_i}\boldsymbol{H}\boldsymbol{e}_i\boldsymbol{e}_i^\top (\sum_{j=m+1}^{n} \sqrt{1-\lambda_j}\boldsymbol{H}\boldsymbol{e}_j\boldsymbol{e}_j^\top)^\top \Big) \\
&= \Big\| \sum_{i=m+1}^{n} \sqrt{1-\lambda_i}\boldsymbol{H}\boldsymbol{e}_i\boldsymbol{e}_i^\top \Big\|_F^2.
\end{aligned}
$$

That is,

$$
\|\boldsymbol{H}\|_E = \Big\| \sum_{i=m+1}^{n} \sqrt{1-\lambda_i}\boldsymbol{H}\boldsymbol{e}_i\boldsymbol{e}_i^\top \Big\|_F.
$$

On the other hand, (3) implies

$$
\|\boldsymbol{H}\|_{\mathcal{M}^\perp} = \|\boldsymbol{H}_{\mathcal{M}^\perp}\|_F = \Big\| \sum_{i=m+1}^{n} \boldsymbol{H}\boldsymbol{e}_i\boldsymbol{e}_i^\top \Big\|_F.
$$

We first show that both $\|\boldsymbol{H}\|_{\mathcal{M}^\perp}$ and $\|\boldsymbol{H}\|_E$ are seminorms. Since $\|c\boldsymbol{H}\|_F = |c| \cdot \|\boldsymbol{H}\|_F$ for any $c \in \mathbb{R}$, we have $\|c\boldsymbol{H}\|_{\mathcal{M}^\perp} = |c| \cdot \|\boldsymbol{H}\|_{\mathcal{M}^\perp}$ and $\|c\boldsymbol{H}\|_E = |c| \cdot \|\boldsymbol{H}\|_E$. Moreover, for any two matrices $\boldsymbol{H}^1$ and $\boldsymbol{H}^2$ s.t. $\boldsymbol{H} = \boldsymbol{H}^1 + \boldsymbol{H}^2$, we have

$$
\sum_{i=m+1}^{n} \boldsymbol{H}^1\boldsymbol{e}_i\boldsymbol{e}_i^\top + \sum_{i=m+1}^{n} \boldsymbol{H}^2\boldsymbol{e}_i\boldsymbol{e}_i^\top = \sum_{i=m+1}^{n} \boldsymbol{H}\boldsymbol{e}_i\boldsymbol{e}_i^\top,
$$

$$
\sum_{i=m+1}^{n} \sqrt{1-\lambda_i}\boldsymbol{H}^1\boldsymbol{e}_i\boldsymbol{e}_i^\top + \sum_{i=m+1}^{n} \sqrt{1-\lambda_i}\boldsymbol{H}^2\boldsymbol{e}_i\boldsymbol{e}_i^\top = \sum_{i=m+1}^{n} \sqrt{1-\lambda_i}\boldsymbol{H}\boldsymbol{e}_i\boldsymbol{e}_i^\top.
$$

505    Then the triangle inequality of $\|\cdot\|_F$ implies that of $\|\boldsymbol{H}\|_{\mathcal{M}^\perp}$ and $\|\boldsymbol{H}\|_E$, respectively.

Now since $0 < 1 - \lambda_{m+1} \leq 1 - \lambda_i \leq 2$ for any $i = m+1, \ldots, n$, we may take $\alpha = \sqrt{1-\lambda_{m+1}}$ and $\beta = \sqrt{2}$. Then

$$
\begin{aligned}
\alpha\|\boldsymbol{H}\|_{\mathcal{M}^\perp} = \Big\|\alpha \sum_{i=m+1}^{n} \boldsymbol{H}\boldsymbol{e}_i\boldsymbol{e}_i^\top \Big\|_F &\leq \Big\| \sum_{i=m+1}^{n} \sqrt{1-\lambda_i}\boldsymbol{H}\boldsymbol{e}_i\boldsymbol{e}_i^\top \Big\|_F \\
&\leq \Big\| \beta \sum_{i=m+1}^{n} \boldsymbol{H}\boldsymbol{e}_i\boldsymbol{e}_i^\top \Big\|_F \\
&= \beta\|\boldsymbol{H}\|_{\mathcal{M}^\perp}.
\end{aligned}
$$

506    The result thus follows from $\|\boldsymbol{H}\|_E = \Big\| \sum_{i=m+1}^{n} \sqrt{1-\lambda_i}\boldsymbol{H}\boldsymbol{e}_i\boldsymbol{e}_i^\top \Big\|_F$.     $\square$

507 **B.1    ReLU**

508    We present a crucial tool to characterize how ReLU affects its input.

**Lemma B.1.** *Let $\boldsymbol{Z} \in \mathbb{R}^{d \times n}$, and let $\boldsymbol{Z}^+ = \max(\boldsymbol{Z}, 0)$ and $\boldsymbol{Z}^- = \max(-\boldsymbol{Z}, 0)$ be the positive and negative parts of $\boldsymbol{Z}$. Then (1) $\boldsymbol{Z}^+, \boldsymbol{Z}^-$ are (component-wise) nonnegative and $\boldsymbol{Z} = \boldsymbol{Z}^+ - \boldsymbol{Z}^-$ and (2) $\langle \boldsymbol{Z}^+, \boldsymbol{Z}^- \rangle_F = 0$.*

*Proof of Lemma B.1.* Notice that for any $a \in \mathbb{R}$, we have

$$\max(a, 0) = \begin{cases} a & \text{if } a \geq 0 \\ 0 & \text{otherwise} \end{cases} \text{ and } \max(-a, 0) = \begin{cases} 0 & \text{if } a \geq 0 \\ -a & \text{otherwise} \end{cases}.$$

This implies that $a = \max(a, 0) - \max(-a, 0)$ and $\max(a, 0) \cdot \max(-a, 0) = 0$.

Let $Z_{ij}$ be the $(i, j)^{th}$ entry of $\boldsymbol{Z}$. Then $\boldsymbol{Z} = \boldsymbol{Z}^+ - \boldsymbol{Z}^-$ follows from $Z_{ij} = \max(Z_{ij}, 0) - \max(-Z_{ij}, 0)$. Also, one can deduce that

$$\langle \boldsymbol{Z}^+, \boldsymbol{Z}^- \rangle_F = \text{Trace}((\boldsymbol{Z}^+)^\top \boldsymbol{Z}^-) = \sum_{i=1}^{d} \sum_{j=1}^{j} \max(Z_{ij}, 0) \max(-Z_{ij}, 0) = 0.$$

$\square$

Before proving Proposition 3.2, we notice the following relation between $\boldsymbol{Z}$ and $\boldsymbol{H}$.

**Lemma B.2.** *Given $\boldsymbol{Z} \in \mathbb{R}^{d \times n}$, let $\boldsymbol{H} = \sigma(\boldsymbol{Z})$ with $\sigma$ being ReLU, then $\boldsymbol{H}$ lies on the high-dimensional sphere, in $\|\cdot\|_F$ norm, that is centered at $\boldsymbol{Z}/2$ and with radius $\|\boldsymbol{Z}/2\|_F$. That is, $\boldsymbol{H}$ and $\boldsymbol{Z}$ satisfy the following equation*

$$\left\| \boldsymbol{H} - \frac{\boldsymbol{Z}}{2} \right\|_F^2 = \left\| \frac{\boldsymbol{Z}}{2} \right\|_F^2. \tag{8}$$

*Proof of Lemma B.2.* We observe that $\boldsymbol{H} = \sigma(\boldsymbol{Z}) = \max(\boldsymbol{Z}, 0) = \boldsymbol{Z}^+$ is the positive part of $\boldsymbol{Z}$. Then

$$\langle \boldsymbol{H}, \boldsymbol{Z} \rangle_F = \langle \boldsymbol{H}, \boldsymbol{Z}^+ - \boldsymbol{Z}^- \rangle_F = \langle \boldsymbol{H}, \boldsymbol{Z}^+ \rangle_F - \langle \boldsymbol{H}, \boldsymbol{Z}^- \rangle_F = \langle \boldsymbol{H}, \boldsymbol{H} \rangle_F,$$

where we have used $\boldsymbol{Z} = \boldsymbol{Z}^+ - \boldsymbol{Z}^-$ and $\langle \boldsymbol{H}, \boldsymbol{Z}^- \rangle_F = \langle \boldsymbol{Z}^+, \boldsymbol{Z}^- \rangle_F = 0$ from Lemma B.1.

Therefore, one can deduce the desired result as follows

$$\langle \boldsymbol{H}, \boldsymbol{H} \rangle_F - \langle \boldsymbol{H}, \boldsymbol{Z} \rangle_F = 0 \Rightarrow \|\boldsymbol{H}\|_F^2 - 2\left\langle \boldsymbol{H}, \frac{\boldsymbol{Z}}{2} \right\rangle_F + \left\| \frac{\boldsymbol{Z}}{2} \right\|_F^2 = \left\| \frac{\boldsymbol{Z}}{2} \right\|_F^2$$

$$\Rightarrow \left\| \boldsymbol{H} - \frac{\boldsymbol{Z}}{2} \right\|_F^2 = \left\| \frac{\boldsymbol{Z}}{2} \right\|_F^2.$$

$\square$

Applying $\|\boldsymbol{H}\|_F^2 = \|\boldsymbol{H}_{\mathcal{M}} + \boldsymbol{H}_{\mathcal{M}^\perp}\|_F^2 = \|\boldsymbol{H}_{\mathcal{M}}\|_F^2 + \|\boldsymbol{H}_{\mathcal{M}^\perp}\|_F^2$, to both $\frac{\boldsymbol{Z}}{2}$ and $\boldsymbol{H} - \frac{\boldsymbol{Z}}{2}$, we obtain

$$\left\| \frac{\boldsymbol{Z}}{2} \right\|_F^2 = \left\| \frac{\boldsymbol{Z}_{\mathcal{M}^\perp}}{2} \right\|_F^2 + \left\| \frac{\boldsymbol{Z}_{\mathcal{M}}}{2} \right\|_F^2,$$

and

$$\left\| \boldsymbol{H} - \frac{\boldsymbol{Z}}{2} \right\|_F^2 = \left\| \boldsymbol{H}_{\mathcal{M}^\perp} - \frac{\boldsymbol{Z}_{\mathcal{M}^\perp}}{2} \right\|_F^2 + \left\| \boldsymbol{H}_{\mathcal{M}} - \frac{\boldsymbol{Z}_{\mathcal{M}}}{2} \right\|_F^2.$$

Then (8) becomes

$$\left\| \frac{\boldsymbol{Z}_{\mathcal{M}^\perp}}{2} \right\|_F^2 - \left\| \boldsymbol{H}_{\mathcal{M}^\perp} - \frac{\boldsymbol{Z}_{\mathcal{M}^\perp}}{2} \right\|_F^2 = \left\| \boldsymbol{H}_{\mathcal{M}} - \frac{\boldsymbol{Z}_{\mathcal{M}}}{2} \right\|_F^2 - \left\| \frac{\boldsymbol{Z}_{\mathcal{M}}}{2} \right\|_F^2 \tag{9}$$

By direct calculation, we have

$$\left\| \boldsymbol{H}_{\mathcal{M}} - \frac{\boldsymbol{Z}_{\mathcal{M}}}{2} \right\|_F^2 - \left\| \frac{\boldsymbol{Z}_{\mathcal{M}}}{2} \right\|_F^2 = \langle \boldsymbol{H}_{\mathcal{M}}, \boldsymbol{H}_{\mathcal{M}} \rangle_F - 2\left\langle \boldsymbol{H}_{\mathcal{M}}, \frac{\boldsymbol{Z}_{\mathcal{M}}}{2} \right\rangle_F$$
$$= \langle \boldsymbol{H}_{\mathcal{M}}, \boldsymbol{H}_{\mathcal{M}} - \boldsymbol{Z}_{\mathcal{M}} \rangle_F. \tag{10}$$

Combining (9) and (10), we obtain the following result

**Lemma B.3.** *For any $\boldsymbol{Z} = \boldsymbol{Z}_{\mathcal{M}} + \boldsymbol{Z}_{\mathcal{M}^\perp}$, let $\boldsymbol{H} = \sigma(\boldsymbol{Z}) = \boldsymbol{H}_{\mathcal{M}} + \boldsymbol{H}_{\mathcal{M}^\perp}$, then*

$$\left\|\frac{\boldsymbol{Z}_{\mathcal{M}^\perp}}{2}\right\|_F^2 - \left\|\boldsymbol{H}_{\mathcal{M}^\perp} - \frac{\boldsymbol{Z}_{\mathcal{M}^\perp}}{2}\right\|_F^2 = \langle \boldsymbol{Z}_{\mathcal{M}}^+, \boldsymbol{Z}_{\mathcal{M}}^- \rangle_F.$$

*where $\boldsymbol{Z}_{\mathcal{M}}^+ = \sum_{i=1}^m \boldsymbol{Z}^+ \boldsymbol{e}_i \boldsymbol{e}_i^\top, \boldsymbol{Z}_{\mathcal{M}}^- = \sum_{i=1}^m \boldsymbol{Z}^- \boldsymbol{e}_i \boldsymbol{e}_i^\top$.*

*Proof of Lemma B.3.* Recall that $\boldsymbol{H} = \sigma(\boldsymbol{Z}) = \max(\boldsymbol{Z}, 0) = \boldsymbol{Z}^+$. Also, $\boldsymbol{Z} = \boldsymbol{Z}^+ - \boldsymbol{Z}^-$ implies $\boldsymbol{Z}_{\mathcal{M}} = \boldsymbol{Z}_{\mathcal{M}}^+ - \boldsymbol{Z}_{\mathcal{M}}^- = \boldsymbol{H}_{\mathcal{M}}^+ - \boldsymbol{Z}_{\mathcal{M}}^-$. Therefore, we see that

$$\langle \boldsymbol{H}_{\mathcal{M}}, \boldsymbol{H}_{\mathcal{M}} - \boldsymbol{Z}_{\mathcal{M}} \rangle_F = \langle \boldsymbol{Z}_{\mathcal{M}}^+, \boldsymbol{Z}_{\mathcal{M}}^- \rangle_F.$$

$\square$

By using the fact that $\langle \boldsymbol{Z}_{\mathcal{M}}^+, \boldsymbol{Z}_{\mathcal{M}}^- \rangle_F \geq 0$ in Lemma B.3, we reveal a geometric relation between $\boldsymbol{Z}$ and $\boldsymbol{H}$ mentioned in Proposition 3.2.

*Proof of Proposition 3.2.* Since $\boldsymbol{Z}^+, \boldsymbol{Z}^- \geq 0$ are nonnegative and all the eigenvectors $\boldsymbol{e}_i$ are also nonnegative, we see that $\boldsymbol{Z}_{\mathcal{M}}^+ = \sum_{i=1}^m \boldsymbol{Z}^+ \boldsymbol{e}_i \boldsymbol{e}_i^\top$ and $\boldsymbol{Z}_{\mathcal{M}}^- = \sum_{i=1}^m \boldsymbol{Z}^- \boldsymbol{e}_i \boldsymbol{e}_i^\top$ are nonnegative. This indicates that

$$\langle \boldsymbol{Z}_{\mathcal{M}}^+, \boldsymbol{Z}_{\mathcal{M}}^- \rangle_F = \mathrm{Trace}\left(\boldsymbol{Z}_{\mathcal{M}}^+ (\boldsymbol{Z}_{\mathcal{M}}^-)^\top\right) \geq 0.$$

Then according to Lemma B.3, we obtain

$$\left\|\frac{\boldsymbol{Z}_{\mathcal{M}^\perp}}{2}\right\|_F^2 - \left\|\boldsymbol{H}_{\mathcal{M}^\perp} - \frac{\boldsymbol{Z}_{\mathcal{M}^\perp}}{2}\right\|_F^2 = \langle \boldsymbol{Z}_{\mathcal{M}}^+, \boldsymbol{Z}_{\mathcal{M}}^- \rangle_F \geq 0.$$

So we have

$$\left\|\boldsymbol{H}_{\mathcal{M}^\perp} - \frac{\boldsymbol{Z}_{\mathcal{M}^\perp}}{2}\right\|_F = \sqrt{\left\|\frac{\boldsymbol{Z}_{\mathcal{M}^\perp}}{2}\right\|_F^2 - \langle \boldsymbol{Z}_{\mathcal{M}}^+, \boldsymbol{Z}_{\mathcal{M}}^- \rangle_F}$$

$$= \sqrt{\left\|\frac{\boldsymbol{Z}_{\mathcal{M}^\perp}}{2}\right\|_F^2 - \langle \boldsymbol{H}_{\mathcal{M}}, \boldsymbol{H}_{\mathcal{M}} - \boldsymbol{Z}_{\mathcal{M}} \rangle_F},$$

which shows that $\boldsymbol{H}_{\mathcal{M}^\perp}$ lies on the high-dimensional sphere that we have claimed. Furthermore, we conclude that

$$0 \leq \left\|\boldsymbol{H}_{\mathcal{M}^\perp} - \frac{\boldsymbol{Z}_{\mathcal{M}^\perp}}{2}\right\|_F \leq \left\|\frac{\boldsymbol{Z}_{\mathcal{M}^\perp}}{2}\right\|_F. \tag{11}$$

This demonstrates that $\boldsymbol{H}_{\mathcal{M}^\perp}$ lies on the high-dimensional sphere we have stated.

Since the sphere $\left\|\boldsymbol{H}_{\mathcal{M}^\perp} - \frac{\boldsymbol{Z}_{\mathcal{M}^\perp}}{2}\right\|_F^2 = \left\|\frac{\boldsymbol{Z}_{\mathcal{M}^\perp}}{2}\right\|_F^2$ passes through the origin, the distance of any $\boldsymbol{H}_{\mathcal{M}^\perp}$ to the origin must be no greater than the diameter of this sphere, i.e., $\|\boldsymbol{H}_{\mathcal{M}^\perp}\|_F \leq \|\boldsymbol{Z}_{\mathcal{M}^\perp}\|_F$. Also, this can be derived from

$$\|\boldsymbol{H}_{\mathcal{M}^\perp}\|_F - \left\|\frac{\boldsymbol{Z}_{\mathcal{M}^\perp}}{2}\right\|_F \leq \left\|\boldsymbol{H}_{\mathcal{M}^\perp} - \frac{\boldsymbol{Z}_{\mathcal{M}^\perp}}{2}\right\|_F \leq \left\|\frac{\boldsymbol{Z}_{\mathcal{M}^\perp}}{2}\right\|_F.$$

One can see that the maximal smoothness $\|\boldsymbol{H}_{\mathcal{M}^\perp}\|_F = \|\boldsymbol{Z}_{\mathcal{M}^\perp}\|_F$ is attained when $\boldsymbol{H}_{\mathcal{M}^\perp} = \boldsymbol{Z}_{\mathcal{M}^\perp}$, the intersection of the surface and the line passing through the center and the origin.

After all, we complete the proof by using the fact that $\|\boldsymbol{Z}_{\mathcal{M}^\perp}\|_F = \|\boldsymbol{Z}\|_{\mathcal{M}^\perp}$ for any matrix $\boldsymbol{Z}$, which implies $\|\boldsymbol{H}\|_{\mathcal{M}^\perp} = \|\boldsymbol{H}_{\mathcal{M}^\perp}\|_F \leq \|\boldsymbol{Z}_{\mathcal{M}^\perp}\|_F = \|\boldsymbol{Z}\|_{\mathcal{M}^\perp}$.

$\square$

## B.2 Leaky ReLU

For the leaky ReLU activation function, we have

**Lemma B.4.** *If $\boldsymbol{H} = \sigma_a(\boldsymbol{Z})$ with $\sigma_a$ being leaky ReLU, then $\boldsymbol{H}$ lies on the high-dimensional sphere centered at $(1 + a)\boldsymbol{Z}/2$ with radius $\|(1 - a)\boldsymbol{Z}/2\|_F$.*

*Proof of Lemma B.4.* Notice that

$$H = \sigma_a(Z) = Z^+ - aZ^-.$$

Then $H - Z = (1-a)Z^-$ and $H - aZ = (1-a)Z^+$. Using $\langle Z^-, Z^+ \rangle_F = 0$, we have

$$\langle H - Z, H - aZ \rangle_F = 0 \Rightarrow \|H\|_F^2 - 2\left\langle H, \frac{(1+a)Z}{2}\right\rangle_F + a\|Z\|_F^2 = 0$$
$$\Rightarrow \|H\|_F^2 - 2\left\langle H, \frac{(1+a)Z}{2}\right\rangle_F = -a\|Z\|_F^2$$
$$\Rightarrow \left\|H - \frac{(1+a)}{2}Z\right\|_F^2 = \left\|\frac{(1+a)}{2}Z\right\|_F^2 - a\|Z\|_F^2 = \left\|\frac{(1-a)}{2}Z\right\|_F^2.$$

$\square$

Moreover, we notice that

**Lemma B.5.** *For any $Z = Z_{\mathcal{M}} + Z_{\mathcal{M}^\perp}$, let $H = \sigma_a(Z) = H_{\mathcal{M}} + H_{\mathcal{M}^\perp}$, then*

$$\left\|\frac{(1-a)}{2}Z_{\mathcal{M}^\perp}\right\|_F^2 - \left\|H_{\mathcal{M}^\perp} - \frac{(1+a)}{2}Z_{\mathcal{M}^\perp}\right\|_F^2 = (1-a)^2\langle Z_{\mathcal{M}}^+, Z_{\mathcal{M}}^-\rangle_F$$

*Proof of Lemma B.5.* Similar to the proof of Lemma B.3, the orthogonal decomposition implies that

$$\left\|\frac{(1-a)}{2}Z_{\mathcal{M}^\perp}\right\|_F^2 - \left\|H_{\mathcal{M}^\perp} - \frac{(1+a)}{2}Z_{\mathcal{M}^\perp}\right\|_F^2 = \left\|H_{\mathcal{M}} - \frac{(1+a)}{2}Z_{\mathcal{M}}\right\|_F^2 - \left\|\frac{(1-a)}{2}Z_{\mathcal{M}}\right\|_F^2$$
$$= \langle H_{\mathcal{M}} - Z_{\mathcal{M}}, H_{\mathcal{M}} - aZ_{\mathcal{M}}\rangle_F$$
$$= \langle (1-a)Z_{\mathcal{M}}^-, (1-a)Z_{\mathcal{M}}^+\rangle_F$$
$$= (1-a)^2\langle Z_{\mathcal{M}}^-, Z_{\mathcal{M}}^+\rangle_F.$$

$\square$

*Proof of Proposition 3.3.* Similar to the proof of Proposition 3.2, we apply $\langle Z_{\mathcal{M}}^-, Z_{\mathcal{M}}^+\rangle_F \geq 0$ to Lemma B.5 and hence obtain the geometric condition as follows

$$\left\|H_{\mathcal{M}^\perp} - \frac{(1+a)}{2}Z_{\mathcal{M}^\perp}\right\|_F = \sqrt{\left\|\frac{(1-a)}{2}Z_{\mathcal{M}^\perp}\right\|_F^2 - \langle H_{\mathcal{M}} - Z_{\mathcal{M}}, H_{\mathcal{M}} - aZ_{\mathcal{M}}\rangle_F}.$$

Then we have the following inequality

$$0 \leq \left\|H_{\mathcal{M}^\perp} - \frac{(1+a)}{2}Z_{\mathcal{M}^\perp}\right\|_F \leq \left\|\frac{(1-a)}{2}Z_{\mathcal{M}^\perp}\right\|_F.$$

Moreover, we deduce that

$$\left| \|H_{\mathcal{M}^\perp}\|_F - \left\|\frac{(1+a)}{2}Z_{\mathcal{M}^\perp}\right\|_F \right| \leq \left\|H_{\mathcal{M}^\perp} - \frac{(1+a)}{2}Z_{\mathcal{M}^\perp}\right\|_F \leq \left\|\frac{(1-a)}{2}Z_{\mathcal{M}^\perp}\right\|_F.$$

and hence

$$-\left\|\frac{(1-a)}{2}Z_{\mathcal{M}^\perp}\right\|_F \leq \|H_{\mathcal{M}^\perp}\|_F - \left\|\frac{(1+a)}{2}Z_{\mathcal{M}^\perp}\right\|_F \leq \left\|\frac{(1-a)}{2}Z_{\mathcal{M}^\perp}\right\|_F.$$

Therefore, we obtain $a\|Z_{\mathcal{M}^\perp}\|_F \leq \|H_{\mathcal{M}^\perp}\|_F \leq \|Z_{\mathcal{M}^\perp}\|_F$. (Remark that $H_{\mathcal{M}^\perp}$ achieves its maximal norm when it is equal to $Z_{\mathcal{M}^\perp}$, the intersection of the surface and the line passing through the center and the origin. )

By using the fact that $\|Z_{\mathcal{M}^\perp}\|_F = \|Z\|_{\mathcal{M}^\perp}$ for any matrix $Z$, we conclude that $a\|Z\|_{\mathcal{M}^\perp} \leq \|H\|_{\mathcal{M}^\perp} \leq \|Z\|_{\mathcal{M}^\perp}$. $\square$

# C   Proofs in Section 4

Throughout this section, we assume that $z_{\mathcal{M}^\perp} \neq \mathbf{0}$.

 *Proof of Proposition 4.3.* Recall that $e = \tilde{D}^{\frac{1}{2}} u_n / c$ has only positive entries where $\tilde{D}$ is the aug-
 mented degree matrix and $u_n = [1, \ldots, 1]^\top \in \mathbb{R}^n$ and $c = \|\tilde{D}^{\frac{1}{2}} u_n\|$. Let $d_i$ be the $i^{th}$ diagonal
 entry of $\tilde{D}$. Then we have $e = [\sqrt{d_1}/c, \sqrt{d_2}/c, \ldots, \sqrt{d_n}/c]^\top$ and $c = \sqrt{\sum_{i=1}^n d_i}$.

Note that $z(\alpha) = z - \alpha e = z - \frac{\alpha}{c}\tilde{D}^{\frac{1}{2}} u_n = \tilde{D}^{\frac{1}{2}}(\tilde{D}^{-\frac{1}{2}} z - \frac{\alpha}{c} u_n) = \tilde{D}^{\frac{1}{2}}(x - \frac{\alpha}{c} u_n)$, where we assume $x := \tilde{D}^{-\frac{1}{2}} z$. Then we observe that when $\sigma$ is the ReLU activation function,

$$h(\alpha) = \sigma(z(\alpha)) = \sigma\left(\tilde{D}^{\frac{1}{2}}\left(x - \frac{\alpha}{c} u_n\right)\right) = \tilde{D}^{\frac{1}{2}}\sigma\left(x - \frac{\alpha}{c} u_n\right),$$

and hence

$$\langle h(\alpha), e \rangle = \left\langle \tilde{D}^{\frac{1}{2}}\sigma\left(x - \frac{\alpha}{c} u_n\right), e \right\rangle$$
$$= \left\langle \sigma\left(x - \frac{\alpha}{c} u_n\right), \tilde{D}^{\frac{1}{2}} e \right\rangle = \left\langle \sigma\left(x - \frac{\alpha}{c} u_n\right), \tilde{D} u_n \right\rangle.$$

We may now assume $x = [x_1, \ldots, x_n]^\top$ is well-ordered s.t. $x_1 \geq x_2 \geq \ldots \geq x_n$. Indeed, there is a collection of indices $\{k_1, \ldots, k_l\}$ s.t.

$$x_1 = \ldots, x_{k_1} \text{ and } x_{k_1} > x_{k_1+1},$$
$$x_{k_{j-1}+1} = \ldots = x_{k_j} \text{ and } x_{k_j} > x_{k_j+1} \text{ for any } j = 2, \ldots, l-1,$$
$$x_{k_{l-1}+1} = \ldots = x_{k_l} \text{ and } k_l = n.$$

 That is, $x_1 = x_2 = \ldots = x_{k_1} > x_{k_1+1} = \ldots = x_{k_2} > x_{k_2+1} = \ldots = x_{k_3} > x_{k_3+1} \ldots$

We first restrict the domain of $\alpha$ s.t. $h(\alpha) \neq 0$. Note that we have

$$h(\alpha) = 0 \Leftrightarrow \sigma\left(x - \frac{\alpha}{c} u_n\right) = 0$$
$$\Leftrightarrow x_i - \frac{\alpha}{c} \leq 0 \text{ for } i = 1, \ldots, n$$
$$\Leftrightarrow x_1 - \frac{\alpha}{c} \leq 0$$
$$\Leftrightarrow \alpha \geq c x_1.$$

 So we will study the smoothness $s(h(\alpha))$ when $\alpha < c x_1$.

Let $\epsilon > 0$ and consider $\alpha = c(x_1 - \epsilon)$. When $\epsilon \leq x_1 - x_{k_1+1} = x_1 - x_{k_2}$, we see that

$$x - \frac{\alpha}{c} u_n = [\epsilon, \ldots, \epsilon, \epsilon - (x_1 - x_{k_1+1}), \ldots, \epsilon - (x_1 - x_n)]^\top,$$

where only the first $k_1$ entries are positive since $x_1 - x_i \geq \epsilon$ for any $i \geq k_1 + 1$. Therefore,

$$h(\alpha) = \tilde{D}^{\frac{1}{2}}\sigma\left(x - \frac{\alpha}{c} u_n\right) = \tilde{D}^{\frac{1}{2}}[\epsilon, \ldots, \epsilon, 0, \ldots, 0]^\top$$
$$= [\epsilon\sqrt{d_1}, \ldots, \epsilon\sqrt{d_{k_1}}, 0, \ldots, 0]^\top.$$

and hence we can compute that $\|h(\alpha)\| = \epsilon\sqrt{\sum_{i=1}^{k_1} d_i}$. Also, we have

$$\|h(\alpha)\|_{\mathcal{M}} = |\langle h(\alpha), e \rangle| = [\epsilon\sqrt{d_1}, \ldots, \epsilon\sqrt{d_{k_1}}, 0, \ldots, 0]^\top [\sqrt{d_1}/c, \sqrt{d_2}/c, \ldots, \sqrt{d_n}/c]$$
$$= \frac{\epsilon}{c}\sum_{i=1}^{k_1} d_i.$$

Then we obtain the smoothness $s(h(\alpha))$ as follows

$$s(h(\alpha)) = \frac{\|h(\alpha)\|_{\mathcal{M}}}{\|h(\alpha)\|} = \frac{\frac{\epsilon}{c}\sum_{i=1}^{k_1} d_i}{\epsilon\sqrt{\sum_{i=1}^{k_1} d_i}} = \frac{\sqrt{\sum_{i=1}^{k_1} d_i}}{c} = \frac{K_1}{c} < 1,$$

557  where $K_1 := \sqrt{\sum_{i=1}^{k_1} d_i}$. Similarly, we may denote $\sqrt{\sum_{i=k_{j-1}+1}^{k_j} d_i}$ by $K_j$ for $j = 2, \ldots, l$.

558  Now we are going to show that the smoothness $s(\boldsymbol{h}(\alpha))$ is increasing as $\alpha$ gets smaller whenever $\alpha <$
559  $cx_1$, implying $\frac{K_1}{c}$ is the minimum of the smoothness $s(\boldsymbol{h}(\alpha))$. Remember that we are considering
560  $\alpha = c(x_1 - \epsilon)$ and we have studied the case when $0 < \epsilon \leq x_1 - x_{k_1+1} = x_1 - x_{k_2}$.

Let $\delta_j := x_1 - x_{k_j}$ for $1 \leq j \leq l$. Clearly, we have $\delta_1 = 0$ and $\delta_j < \delta_{j+1}$ for $1 \leq j \leq l - 1$. Fix a $j' \in \{2, \ldots, l-1\}$, we see that when $\delta_{j'} < \epsilon \leq x_1 - x_{k_{j'}+1}$,

$$\boldsymbol{x} - \frac{\alpha}{c}\boldsymbol{u}_n$$

$$= \left[\epsilon - \delta_1, \ldots, \epsilon - \delta_1, \epsilon - \delta_2, \ldots, \epsilon - \delta_2, \epsilon - \delta_3, \ldots, \epsilon - \delta_{j'}, \epsilon - (x_1 - x_{k_{j'}+1}), \ldots, \epsilon - (x_1 - x_n)\right]^\top,$$

where we have $\epsilon - \delta_j > 0$ for $2 \leq j \leq j'$ and $\epsilon - (x_1 - x_i) \leq 0$ for any $i \geq k_{j'} + 1$. Consequently,

$$\boldsymbol{h}(\alpha) = \tilde{\boldsymbol{D}}^{\frac{1}{2}}\sigma(\boldsymbol{x} - \frac{\alpha}{c}\boldsymbol{u}_n) = [(\epsilon - \delta_1)\sqrt{d_1}, \ldots, (\epsilon - \delta_1)\sqrt{d_{k_1}}, (\epsilon - \delta_2)\sqrt{d_{k_1+1}}, \ldots, (\epsilon - \delta_2)\sqrt{d_{k_2}},$$

$$(\epsilon - \delta_3)\sqrt{d_{k_2+1}}, \ldots, (\epsilon - \delta_{j'})\sqrt{d_{k_{j'}}}, 0, \ldots, 0]^\top.$$

Then we can compute

$$\|\boldsymbol{h}(\alpha)\| = \sqrt{\sum_{j=1}^{j'}\sum_{i=k_{j-1}+1}^{k_j} d_i(\epsilon - \delta_j)^2} = \sqrt{\sum_{j=1}^{j'} K_j^2(\epsilon - \delta_j)^2},$$

where we set $k_0 := 0$ for simplicity and $K_j = \sqrt{\sum_{i=k_{j-1}+1}^{k_j} d_i}$ for $j = 1, \ldots, j'$. Also, we have

$$\|\boldsymbol{h}(\alpha)\|_{\mathcal{M}} = |\langle \boldsymbol{h}(\alpha), \boldsymbol{e}\rangle| = \sum_{j=1}^{j'}\sum_{i=k_{j-1}+1}^{k_j} \frac{d_i(\epsilon - \delta_j)}{c} = \frac{1}{c}\sum_{j=1}^{j'} K_j^2(\epsilon - \delta_j).$$

A careful calculation shows that $\frac{\partial}{\partial \epsilon}s(\boldsymbol{h}(\alpha)) > 0$ whenever $\delta_{j'} < \epsilon \leq x_1 - x_{k_{j'}+1}$ which implies that $s(\boldsymbol{h}(\alpha))$ is increasing as $\epsilon$ increases. Indeed, we have

$$\frac{\partial}{\partial \epsilon}s(\boldsymbol{h}(\alpha))$$

$$= \frac{\partial}{\partial \epsilon}\left(\frac{\sum_{j=1}^{j'} K_j^2(\epsilon - \delta_j)}{c\sqrt{\sum_{j=1}^{j'} K_j^2(\epsilon - \delta_j)^2}}\right)$$

$$= \frac{\left(\frac{\partial}{\partial \epsilon}\sum_{j=1}^{j'} K_j^2(\epsilon - \delta_j)\right)\sqrt{\sum_{j=1}^{j'} K_j^2(\epsilon - \delta_j)^2} - \sum_{j=1}^{j'} K_j^2(\epsilon - \delta_j)\left(\frac{\partial}{\partial \epsilon}\sqrt{\sum_{j=1}^{j'} K_j^2(\epsilon - \delta_j)^2}\right)}{c\sum_{j=1}^{j'} K_j^2(\epsilon - \delta_j)^2}$$

$$= \frac{\left(\sum_{j=1}^{j'} K_j^2\right)\sqrt{\sum_{j=1}^{j'} K_j^2(\epsilon - \delta_j)^2} - \sum_{j=1}^{j'} K_j^2(\epsilon - \delta_j)\left(\frac{\frac{\partial}{\partial \epsilon}\sum_{j=1}^{j'} K_j^2(\epsilon-\delta_j)^2}{2\sqrt{\sum_{j=1}^{j'} K_j^2(\epsilon-\delta_j)^2}}\right)}{c\sum_{j=1}^{j'} K_j^2(\epsilon - \delta_j)^2}$$

$$= \frac{\left(\sum_{j=1}^{j'} K_j^2\right)\sum_{j=1}^{j'} K_j^2(\epsilon - \delta_j)^2 - \sum_{j=1}^{j'} K_j^2(\epsilon - \delta_j)\left(\sum_{j=1}^{j'} K_j^2(\epsilon - \delta_j)\right)}{c\sum_{j=1}^{j'} K_j^2(\epsilon - \delta_j)^2\sqrt{\sum_{j=1}^{j'} K_j^2(\epsilon - \delta_j)^2}}.$$

Then to show that $\frac{\partial}{\partial \epsilon}s(\boldsymbol{h}(\alpha)) > 0$, it suffices to show that the numerator is positive, i.e.

$$\left(\sum_{j=1}^{j'} K_j^2\right)\sum_{j=1}^{j'} K_j^2(\epsilon - \delta_j)^2 - \left(\sum_{j=1}^{j'} K_j^2(\epsilon - \delta_j)\right)^2 > 0,$$

since the denominator $c \sum_{j=1}^{j'} K_j^2 (\epsilon - \delta_j)^2 \sqrt{\sum_{j=1}^{j'} K_j^2 (\epsilon - \delta_j)^2} > 0$ is always positive. In fact, this follows from the Cauchy inequality $\|\boldsymbol{v}\| \|\boldsymbol{u}\| \geq \langle \boldsymbol{v}, \boldsymbol{u} \rangle$, where we set

$$\boldsymbol{v} := [K_1, K_2, \ldots, K_{J'}]^\top, \quad \boldsymbol{u} := [K_1(\epsilon - \delta_1), K_2(\epsilon - \delta_2), \ldots, K_{j'}(\epsilon - \delta_{j'})]^\top.$$

Moreover, equality happens only when $\boldsymbol{v}$ is parallel to $\boldsymbol{u}$. This is, however, impossible since $\epsilon - \delta_j > \epsilon - \delta_{j+1}$ for any $j = 1, \ldots, j' - 1$ and each $K_j$ is positive.

So we see that $s(\boldsymbol{h}(\alpha))$ is increasing as $\epsilon$ increases whenever $0 < \epsilon$, and hence the smoothness $s(\boldsymbol{h}(\alpha))$ is increasing as $\alpha$ decreases whenever $cx_n \leq \alpha < cx_1$.

For the case $j' = l$ where $\delta_l = x_1 - x_n < \epsilon$, we have $x_n - \alpha/c = x_n - (x_1 - \epsilon) = \epsilon - (x_1 - x_n) > 0$, implying $\alpha < cx_n$ and $\boldsymbol{h}(\alpha) = \boldsymbol{z}(\alpha)$. We have shown that the smoothness is increasing as $\alpha$ is going far from $\langle \boldsymbol{z}, \boldsymbol{e} \rangle$; in particular, when $\alpha < \langle \boldsymbol{z}, \boldsymbol{e} \rangle$ and $\alpha$ is decreasing. One can check that

$$cx_n = \frac{\sum_{i=1}^n d_i x_n}{c} = \left\langle x_n \boldsymbol{u}_n, \frac{\tilde{\boldsymbol{D}} \boldsymbol{u}_n}{c} \right\rangle \leq \left\langle \boldsymbol{x}, \frac{\tilde{\boldsymbol{D}} \boldsymbol{u}_n}{c} \right\rangle = \left\langle \tilde{\boldsymbol{D}}^{\frac{1}{2}} \boldsymbol{x}, \frac{\tilde{\boldsymbol{D}}^{\frac{1}{2}} \boldsymbol{u}_n}{c} \right\rangle = \langle \boldsymbol{z}, \boldsymbol{e} \rangle,$$

which means the smoothness is increasing as $\alpha$ decreases whenever $\alpha < cx_n$.

We conclude that the smoothness increases as $\alpha$ decreases provided $\alpha < cx_1$. Also, we have $\sup_{\alpha < cx_1} s(\boldsymbol{h}(\alpha)) = 1$ as the case in the proof of Proposition C.1. One can check that $s(\boldsymbol{h}(\alpha))$ is a continuous function for $\alpha < cx_1$ and thus it has range $[K_1/c, 1)$ by the mean value theorem.

Finally, we can establish the result: $K_1/c = \sqrt{\frac{\sum_{x_i = \max \boldsymbol{x}} d_i}{\sum_{j=1}^n d_j}}$ is the minimum of $s(\boldsymbol{h}(\alpha))$ and 1 is the

maximum of $s(\boldsymbol{h}(\alpha))$ occurring whenever $\alpha \geq cx_1 = \sqrt{\sum_{j=1}^n d_j} \max_i x_i$. Moreover, $s(\boldsymbol{h}(\alpha))$ has

a monotone property when $\alpha < \sqrt{\sum_{j=1}^n d_j} \max_i x_i$ and has range $\left[ \sqrt{\frac{\sum_{x_i = \max \boldsymbol{x}} d_i}{\sum_{j=1}^n d_j}}, 1 \right]$.

It is clear that the assumption on the ordering of the entries of $\boldsymbol{x}$ will not affect this result. $\qquad \square$

To prove Proposition 4.4, we first prove an analogous result for the identity function, that is, $\boldsymbol{h} = \sigma(\boldsymbol{z}) = \boldsymbol{z}$.

**Proposition C.1.** *Suppose $\boldsymbol{z}_{\mathcal{M}^\perp} \neq \boldsymbol{0}$, then $s(\boldsymbol{z}(\alpha))$ achieves its minimum $0$ if $\alpha = \langle \boldsymbol{z}, \boldsymbol{e} \rangle$. Moreover, $\sup_\alpha s(\boldsymbol{z}(\alpha)) = 1$ where $s(\boldsymbol{z}(\alpha))$ is close to $1$ when $\alpha$ is far away from $\langle \boldsymbol{z}, \boldsymbol{e} \rangle$.*

Notice that Proposition C.1 does not consider the activation function.

*Proof of Proposition C.1.* We know that $0 \leq s(\boldsymbol{z}(\alpha)) \leq 1$ and

$$s(\boldsymbol{z}(\alpha)) = \sqrt{1 - \frac{\|\boldsymbol{z}_{\mathcal{M}^\perp}\|^2}{\|\boldsymbol{z}(\alpha)\|^2}} = \sqrt{1 - \frac{\|\boldsymbol{z}_{\mathcal{M}^\perp}\|^2}{\|\boldsymbol{z}_{\mathcal{M}^\perp}\|^2 + \|\boldsymbol{z}(\alpha)_{\mathcal{M}}\|^2}}$$

$$= \sqrt{1 - \frac{\|\boldsymbol{z}_{\mathcal{M}^\perp}\|^2}{\|\boldsymbol{z}_{\mathcal{M}^\perp}\|^2 + \|\boldsymbol{z}_{\mathcal{M}} - \alpha \boldsymbol{e}\|^2}}.$$

Suppose $s(\boldsymbol{z}(\alpha)) = 1$. Then we have $\frac{\|\boldsymbol{z}_{\mathcal{M}^\perp}\|^2}{\|\boldsymbol{z}_{\mathcal{M}^\perp}\|^2 + \|\boldsymbol{z}_{\mathcal{M}} - \alpha \boldsymbol{e}\|^2} = 0$ which forces $\|\boldsymbol{z}_{\mathcal{M}^\perp}\| = 0$. However, this contradicts the hypothesis $\boldsymbol{z}_{\mathcal{M}^\perp} \neq \boldsymbol{0}$. So $s(\boldsymbol{z}(\alpha))$ cannot attain its maximum.

But for any $0 \leq t < 1$, one can see that $s(\boldsymbol{z}(\alpha)) = t$ if and only if

$$\sqrt{1 - \frac{\|\boldsymbol{z}_{\mathcal{M}^\perp}\|^2}{\|\boldsymbol{z}_{\mathcal{M}^\perp}\|^2 + \|\boldsymbol{z}_{\mathcal{M}} - \alpha \boldsymbol{e}\|^2}} = t \Leftrightarrow \frac{\|\boldsymbol{z}_{\mathcal{M}^\perp}\|^2}{\|\boldsymbol{z}_{\mathcal{M}^\perp}\|^2 + \|\boldsymbol{z}_{\mathcal{M}} - \alpha \boldsymbol{e}\|^2} = 1 - t^2$$

$$\Leftrightarrow \|\boldsymbol{z}_{\mathcal{M}^\perp}\|^2 = (1 - t^2)\left(\|\boldsymbol{z}_{\mathcal{M}^\perp}\|^2 + \|\boldsymbol{z}_{\mathcal{M}} - \alpha \boldsymbol{e}\|^2\right)$$

$$\Leftrightarrow t^2 \|\boldsymbol{z}_{\mathcal{M}^\perp}\|^2 = (1 - t^2)\|\boldsymbol{z}_{\mathcal{M}} - \alpha \boldsymbol{e}\|^2$$

$$\Leftrightarrow \|\boldsymbol{z}_{\mathcal{M}} - \alpha \boldsymbol{e}\| = \sqrt{\frac{t^2}{1 - t^2}} \cdot \|\boldsymbol{z}_{\mathcal{M}^\perp}\|$$

This implies that $\sup_\alpha s(\boldsymbol{z}(\alpha)) = 1$ and $s(\boldsymbol{z}(\alpha))$ achieves its minimum $0$ if and only if $\alpha = \langle \boldsymbol{z}, \boldsymbol{e} \rangle$. It is clear that $s(\boldsymbol{z}(\alpha))$ get closer to $1$ when $\alpha$ is going far away from $\langle \boldsymbol{z}, \boldsymbol{e} \rangle$. i.e., $|\alpha - \langle \boldsymbol{z}, \boldsymbol{e} \rangle| = \|\boldsymbol{z}_{\mathcal{M}} - \alpha \boldsymbol{e}\|$ is increasing. $\hfill\square$

*Proof of Proposition 4.4.* First, we notice that leaky ReLU has the following two properties

1. $\sigma_a(x) > 0$ for $x \gg 0$ and $\sigma_a(x) < 0$ for $x \ll 0$.

2. $\sigma_a$ is a non-trivial linear map for $x \gg 0$.

We will use Property 1 to show that $\min_\alpha s(\boldsymbol{h}(\alpha)) = 0$ and Property 2 to show that $\sup_\alpha s(\boldsymbol{h}(\alpha)) = 1$. Notice that $\sigma_a(x) < 0$ for $x \ll 0$ implies that there exists a sufficient small $\alpha_2 < 0$ s.t. all of the entries of $\boldsymbol{h}(\alpha_2)$ are negative and hence $|\langle \boldsymbol{h}(\alpha_2), \boldsymbol{e} \rangle| < 0$. Similarly, $\sigma_a(x) > 0$ for $x \gg 0$ implies that there exists a sufficient large $\alpha_1 > 0$ s.t. all of the entries of $\boldsymbol{h}(\alpha_1)$ are positive and hence $|\langle \boldsymbol{h}(\alpha_1), \boldsymbol{e} \rangle| > 0$. Since $|\langle \boldsymbol{h}(\alpha), \boldsymbol{e} \rangle|$ is a continuous function of $\alpha$ on $[\alpha_1, \alpha_2]$, the Intermediate Value Theorem follows that there exists an $\alpha \in (\alpha_1, \alpha_2)$ s.t. $|\langle \boldsymbol{h}(\alpha), \boldsymbol{e} \rangle| = 0$. Thus by definition $s(\boldsymbol{h}(\alpha)) = |\langle \boldsymbol{h}(\alpha), \boldsymbol{e} \rangle| / \|\boldsymbol{h}(\alpha)\|$, we see that $\min_\alpha s(\boldsymbol{h}(\alpha)) = 0$.

On the other hand, since $\sigma_a$ is a non-trivial linear map for $x \gg 0$, we may assume $\sigma_a(x) = cx$ for $x > x_0$ where $c \neq 0$ is some non-zero constant and $x_0 > 0$ is some positive constant. Then we can choose an $\alpha_0 > \langle \boldsymbol{z}, \boldsymbol{e} \rangle$ s.t. for any $\alpha \geq \alpha_0$, all of the entries of $\boldsymbol{z}(\alpha)$ are greater than $x_0$. Then whenever $\alpha \geq \alpha_0$, we have $\boldsymbol{h}(\alpha) = \sigma_a(\boldsymbol{z}(\alpha)) = c\boldsymbol{z}(\alpha)$. This implies

$$s(\boldsymbol{h}(\alpha)) = \frac{|\langle \boldsymbol{h}(\alpha), \boldsymbol{e} \rangle|}{\|\boldsymbol{h}(\alpha)\|} = \frac{|\langle c\boldsymbol{z}(\alpha), \boldsymbol{e} \rangle|}{\|c\boldsymbol{z}(\alpha)\|} = \frac{|\langle \boldsymbol{z}(\alpha), \boldsymbol{e} \rangle|}{\|\boldsymbol{z}(\alpha)\|} = s(\boldsymbol{z}(\alpha)).$$

Thus $\sup_\alpha s(\boldsymbol{h}(\alpha)) = 1$ follows from the Proof of Proposition C.1 where we see that $\sup_\alpha s(\boldsymbol{z}(\alpha)) = 1$ since $s(\boldsymbol{z}(\alpha))$ gets closer to $1$ as $\alpha$ increases.

$\hfill\square$

*Remark* C.2. Indeed, it holds for any continuous function $f : \mathbb{R} \to \mathbb{R}$ satisfying the following

1. $f(x) > 0$ for $x \gg 0$, $f(x) < 0$ for $x \ll 0$ or $f(x) < 0$ for $x \gg 0$, $f(x) > 0$ for $x \ll 0$,

2. $f$ is a non-trivial linear map for $x \gg 0$ or $x \ll 0$.

One can check the proof above only depends on these two properties. It is worth mentioning that most activation functions, e.g. leaky LU, SiLU, $\tanh$, satisfy condition 1.

*Proof of Corollary 4.5.* For any $\alpha$, we notice that $\|\boldsymbol{z}\|_{\mathcal{M}^\perp} = \|\boldsymbol{z}_{\mathcal{M}^\perp}\|_F = \|\boldsymbol{z}(\alpha)\|_{\mathcal{M}^\perp}$ since $\alpha$ only changes the component of $\boldsymbol{z}$ in the eigenspace $\mathcal{M}$. Also, Propositions 3.2 and 3.3 show that $\|\boldsymbol{z}(\alpha)\|_{\mathcal{M}^\perp} \geq \|\boldsymbol{h}(\alpha)\|_{\mathcal{M}^\perp}$ whenever $\boldsymbol{h}(\alpha) = \sigma(\boldsymbol{z}(\alpha))$ or $\sigma_a(\boldsymbol{z}(\alpha))$. Therefore, we see that $\|\boldsymbol{z}\|_{\mathcal{M}^\perp} \geq \|\boldsymbol{h}(\alpha)\|_{\mathcal{M}^\perp}$ holds for any $\alpha$. Since $\boldsymbol{z}_{\mathcal{M}^\perp} \neq 0$, $s(\boldsymbol{z})$ must lie in $[0, 1)$.

$\hfill\square$

# D  Experimental Details

This part includes the missing details about experimental configurations and additional experimental results for Section 6. All tasks we run using Nvidia RTX 3090, GV100, and Tesla T4 GPUs. All computational performance metrics, including timing procedures, are run using Tesla T4 GPUs from Google Colab.

## D.1  Dataset details

In this section, we briefly describe the benchmark datasets used. Table 3 provides additional details about the underlying graph representation.

**Citation Datasets:** The five citation datasets considered are Cora, Citeseer PubMed, Coauthor-Physics, and Ogbn-arxiv. Each dataset is represented by a graph with nodes representing academic publications, features encoding a bag-of-words description, labels classifying the publication type, and edges representing citations.

618  **Web Knowledge-Base Datasets:** The three web knowledge-base datasets are Cornell, Texas, and
619  Wisconsin. Each dataset is represented by a graph with nodes representing CS department webpages,
620  features encoding a bag-of-words description, edges representing hyper-link connections, and labels
621  classifying the webpage type.

622  **Wikipedia Network Datasets:** The two Wikipedia network datasets are Chameleon and Squirrel.
623  Each dataset is represented by a graph with nodes representing CS department webpages, features en-
624  coding a bag-of-words description, edges representing hyper-link connections, and labels classifying
625  the webpage type.

| | # Nodes | # Edges | # Features | # Classes | Splits (Train/Val/Test) |
|---|---|---|---|---|---|
| Cornell | 183 | 295 | 1,703 | 5 | 48/32/20% |
| Texas | 181 | 309 | 1,703 | 5 | 48/32/20% |
| Wisconsin | 251 | 499 | 1,703 | 5 | 48/32/20% |
| Chameleon | 2,277 | 36,101 | 2,325 | 5 | 48/32/20% |
| Squirrel | 5,201 | 217,073 | 2,089 | 5 | 48/32/20% |
| Citeseer | 3,727 | 4,732 | 3,703 | 6 | 120/500/1000 |
| Cora | 2,708 | 5,429 | 1,433 | 7 | 140/500/1000 |
| PubMed | 19,717 | 44,338 | 500 | 3 | 60/500/1000 |
| Coauthor-Physics | 34,493 | 247,962 | 8415 | 5 | 100/150/34,243 |
| Ogbn-arxiv | 169,343 | 1,166,243 | 128 | 40 | 90,941/29,799/48,603 |

Table 3: Graph statistics.

## D.2 Model size and computational time for citation datasets

627  Table 4 compares the model size and computational time for experiments on citation datasets in
628  Section 6.2.

| | # Parameters | Training Time (s) | Inference Time (ms) |
|---|---|---|---|
| **Cora** | | | |
| GCN | 100,423 | 8.4 | 1.6 |
| GCNII | 110,535 | 10.0 | 2.1 |
| GCNII | 708,743 | 57.6 | 12.3 |
| GCNII-SCT | 1,237,127 | 110.3 | 29.6 |
| EGNN | 712,839 | 65.6 | 14.4 |
| EGNN-SCT | 316,551 | 24.8 | 4.5 |
| **Citeseer** | | | |
| GCN | 245,638 | 8.3 | 1.5 |
| GCN-SCT | 301,830 | 15.5 | 4.0 |
| GCNII | 999,174 | 57.6 | 12.3 |
| GCNII-SCT | 1,001,222 | 65.9 | 15.7 |
| EGNN | 739,078 | 39.6 | 7.2 |
| EGNN-SCT | 540,934 | 24.0 | 5.8 |
| **PubMed** | | | |
| GCN | 40,451 | 9.0 | 1.8 |
| GCN-SCT | 40,707 | 11.1 | 2.2 |
| GCNII | 326,659 | 98.2 | 12.8 |
| GCNII-SCT | 590,851 | 71.7 | 17.4 |
| EGNN | 592,899 | 93.7 | 2.5 |
| EGNN-SCT | 130,563 | 16.0 | 3.1 |
| **Coauthor-Physics** | | | |
| GCN | 547,141 | 35.2 | 8.0 |
| GCN-SCT | 547,397 | 33.9 | 8.3 |
| GCNII | 555,333 | 49.1 | 10.3 |
| GCNII-SCT | 555,461 | 67.0 | 9.5 |
| EGNN | 672,069 | 176.4 | 47.9 |
| EGNN-SCT | 572,229 | 51.7 | 14.8 |
| **Ogbn-arxiv** | | | |
| GCN | 27,240 | 50.4 | 21.1 |
| GCN-SCT | 28,392 | 62.6 | 24.4 |
| GCNII | 76,392 | 205.4 | 94.8 |
| GCNII-SCT | 80,616 | 253.0 | 108.9 |
| EGNN | 77,416 | 206.8 | 98.0 |
| EGNN-SCT | 81,640 | 254.0 | 112.3 |

Table 4: Number of model parameters for varying numbers of layers using the optimal model hyperparameters. The SCT is added at each layer and the size of the additional parameters scales with the number of eigenvectors with an eigenvalue of one for matrix $G$ in (2).

 **D.3 Additional Section 6.2 details for citation datasets**

Table 5 lists the hyperparameters used in the grid search in generating the results in Table 1. Also, Table 7 reports the classification accuracy of different models with different depths using either ReLU or leaky ReLU.

| Parameter | Values |
|---|---|
| Learning Rate | $\{1e\text{-}4, 1e\text{-}3, 1e\text{-}2\}$ |
| Weight Decay (FC) | $\{0, 1e\text{-}4, 5e\text{-}4, 1e\text{-}3, 5e\text{-}3, 1e\text{-}2\}$ |
| Weight Decay (Conv) | $\{0, 1e\text{-}4, 5e\text{-}4, 1e\text{-}3, 5e\text{-}3, 1e\text{-}2\}$ |
| Dropout | $\{0.1, 0.2, 0.3, 0.4, 0.5, 0.6, 0.7, 0.8, 0.9\}$ |
| Hidden Channels | $\{16, 32, 64, 128\}$ |
| GCNII-$\alpha$ | $\{0.1, 0.2, 0.3, 0.4, 0.5, 0.6, 0.7, 0.8, 0.9\}$ |
| GCNII-$\theta$ | $\{0.1, 0.2, 0.3, 0.4, 0.5, 0.6, 0.7, 0.8, 0.9\}$ |
| EGNN-$c_{\max}$ | $\{0.5, 1.0, 1.5, 2.0\}$ |
| EGNN-$\alpha$ | $\{0.1, 0.2, 0.3, 0.4, 0.5, 0.6, 0.7, 0.8, 0.9\}$ |
| EGNN-$\theta$ | $\{0.1, 0.2, 0.3, 0.4, 0.5, 0.6, 0.7, 0.8, 0.9\}$ |

Table 5: Hyperparameter grid search for Table 1.

| Layers | 2 | 4 | 16 | 32 |
|---|---|---|---|---|
| **Cora** | | | | |
| EGNN/EGNN-SCT | 83.2/**83.4** | 84.2/**84.3** | 85.4/**85.5** | 85.3/**85.5** |
| **Citeseer** | | | | |
| EGNN/EGNN-SCT | 72.0/**72.1** | 71.9/**72.3** | 72.4/**72.6** | 72.3/**72.8** |
| **PubMed** | | | | |
| EGNN/EGNN-SCT | 79.2/**79.4** | 79.5/**79.8** | **80.1/80.1** | 80.0/**80.2** |
| **Coauthor-Physics** | | | | |
| EGNN/EGNN-SCT | 92.6/**92.8** | 92.9/**93.0** | 93.1/**93.3** | **93.3/93.3** |
| **Ogbn-arxiv** | | | | |
| EGNN/EGNN-SCT | 68.4/**68.5** | 71.1/**71.3** | 72.7/**73.0** | 72.7/**72.9** |

Table 6: Test accuracy for EGNN and EGNN-SCT using SReLU activation function of varying depth on citation networks with the split discussed in Section 6.2. (Unit:%)

**D.3.1 Vanishing gradients**

Figure 4 shows the vanishing gradient problem for training deep GCN – with or without SCT – in comparison to models like GCNII and EGNN. This figure plots $||\partial \boldsymbol{H}^{\text{out}}/\partial \boldsymbol{H}^l||$ for layers $l \in [0, 32]$ as the training epochs run from 0 to 100. Figures 4 (a) and (b) illustrate the vanishing gradient issue for GCN and that it persists for GCN-SCT. Figures 4 (c) and (e) illustrate that GCNII and EGNN do not suffer from vanishing gradients, and furthermore, because these models connect $\boldsymbol{H}^0$ to every layer, the gradient with respect to the weights in the first layer is nonzero. What is interesting about the addition of SCT to both EGNN and GCNII is that the intermediate gradients become large as the training epochs progress shown in Figure 4 (d) and (f).

**D.4 Additional Section 6.2 details for other datasets**

Table 8 reports the mean test accuracy and standard deviation over ten folds of the WebKB and WikipediaNetwork datasets using SCT-based models.

Table 9 lists the average computational time for each epoch for different models of the same depth – 8 layers. These results show that integrating SCT into GNNs only results in a small amount of computational overhead.

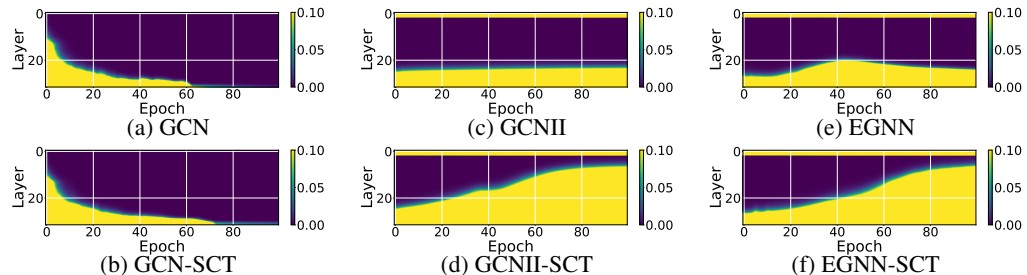

Figure 4: Training gradients for $||\partial \boldsymbol{H}^{\text{out}}/\partial \boldsymbol{H}^l||$ for $l \in [0, 32]$ layers and 100 training epochs on the Citeseer dataset. Here, all models have 32 layers and 16 hidden dimensions for each layer. We observe that (a) GCN suffers from vanishing gradients. By contrast (c) GCNII and (e) EGNN do not suffer from vanishing gradients, and we can observe their skip connection to $\boldsymbol{H}^0$. Because these models (GCNII/GCNII-SCT and EGNN/EGNN-SCT) connect $\boldsymbol{H}^0$ to every layer, the gradient at the first layer is nonzero. We notice that while SCT does not overcome vanishing gradients for (b) GCN-SCT, it is able to increase the norm of the gradients for the intermediate layers in (d) GCNII-SCT and (f) EGNN-SCT.

| Cora | | | | | | | | |
|---|---|---|---|---|---|---|---|---|
| | ReLU | | | | leaky ReLU | | | |
| Layers | 2 | 4 | 16 | 32 | 2 | 4 | 16 | 32 |
| GCN-SCT | 81.2 | 80.3 | 71.4 | 67.2 | 82.9 | 82.8 | 68.0 | 65.5 |
| GCNII-SCT | 83.5 | 83.8 | 82.7 | 83.3 | 83.8 | 84.8 | 84.8 | 85.5 |
| EGNN-SCT | 84.1 | 83.8 | 82.3 | 80.8 | 83.7 | 84.5 | 83.3 | 82.0 |
| Citeseer | | | | | | | | |
| | ReLU | | | | leaky ReLU | | | |
| Layers | 2 | 4 | 16 | 32 | 2 | 4 | 16 | 32 |
| GCN-SCT | 69.0 | 67.3 | 51.5 | 50.3 | 69.9 | 67.7 | 55.4 | 51.0 |
| GCNII-SCT | 72.8 | 72.8 | 72.8 | 73.3 | 72.8 | 72.9 | 73.8 | 72.7 |
| EGNN-SCT | 72.5 | 72.0 | 70.2 | 71.8 | 73.1 | 71.7 | 72.6 | 72.9 |
| PubMed | | | | | | | | |
| | ReLU | | | | leaky ReLU | | | |
| Layers | 2 | 4 | 16 | 32 | 2 | 4 | 16 | 32 |
| GCN-SCT | 79.4 | 78.2 | 75.9 | 77.0 | 79.8 | 78.4 | 76.1 | 76.9 |
| GCNII-SCT | 79.7 | 80.1 | 80.7 | 80.7 | 79.6 | 80.0 | 80.3 | 80.7 |
| EGNN-SCT | 79.7 | 80.1 | 80.0 | 80.4 | 79.8 | 80.4 | 80.3 | 80.2 |
| Coauthor-Physics | | | | | | | | |
| | ReLU | | | | leaky ReLU | | | |
| Layers | 2 | 4 | 16 | 32 | 2 | 4 | 16 | 32 |
| GCN-SCT | 91.8 ± 1.6 | 91.6 ± 3.0 | 44.5 ± 13.0 | 42.6 ± 17.0 | 92.6 ± 1.6 | 92.5 ± 5.9 | 50.9 ± 15.0 | 43.6 ± 16.0 |
| GCNII-SCT | 94.4 ± 0.4 | 93.5 ± 1.2 | 93.7 ± 0.7 | 93.8 ± 0.6 | 94.0 ± 0.4 | 94.2 ± 0.3 | 93.3 ± 0.7 | 94.1 ± 0.3 |
| EGNN-SCT | 93.6 ± 0.7 | 94.1 ± 0.4 | 93.4 ± 0.8 | 93.8 ± 1.3 | 93.9 ± 0.7 | 94.0 ± 0.7 | 94.0 ± 0.7 | 93.3 ± 0.9 |
| Ogbn-arxiv | | | | | | | | |
| | ReLU | | | | leaky ReLU | | | |
| Layers | 2 | 4 | 16 | 32 | 2 | 4 | 16 | 32 |
| GCN-SCT | 71.7 ± 0.3 | 72.6 ± 0.3 | 71.4 ± 0.2 | 71.9 ± 0.3 | 72.1 ± 0.3 | 72.7 ± 0.3 | 72.3 ± 0.2 | 72.3 ± 0.3 |
| GCNII-SCT | 71.4 ± 0.3 | 72.1 ± 0.3 | 72.2 ± 0.2 | 71.8 ± 0.2 | 72.0 ± 0.3 | 72.2 ± 0.2 | 72.4 ± 0.3 | 72.1 ± 0.3 |
| EGNN-SCT | 68.5 ± 0.6 | 71.0 ± 0.5 | 72.8 ± 0.5 | 72.1 ± 0.6 | 67.7 ± 0.5 | 71.3 ± 0.5 | 72.3 ± 0.5 | 72.3 ± 0.5 |

Table 7: Test accuracy results for models of varying depth with ReLU or leaky ReLU activation function on the citation network datasets using the split discussed in Section 6.2.

| | Cornell | Texas | Wisconsin | Chameleon | Squirrel |
|---|---|---|---|---|---|
| GCN-SCT | 55.95 ± 8.5 | 62.16 ± 5.7 | 54.71 ± 4.4 | 38.44 ± 4.3 | 35.31 ± 1.9 |
| GCNII-SCT | 75.41 ± 2.2 | 83.34 ± 4.5 | 86.08 ± 3.8 | 64.52 ± 2.2 | 47.51 ± 1.4 |

Table 8: Test mean ± standard deviation accuracy from 10 fold cross validation on five heterophilic datasets with fixed $48/32/20\%$ splits. The depth of each model is 8 layers with 16 hidden channels. (Unit: second)

| | Cornell | Texas | Wisconsin | Chameleon | Squirrel |
|---|---|---|---|---|---|
| GCN [20] | 0.011 | 0.013 | 0.012 | 0.011 | 0.022 |
| GCNII [6] | 0.017 | 0.018 | 0.017 | 0.013 | 0.022 |
| GCN-SCT | 0.015 | 0.017 | 0.015 | 0.011 | 0.023 |
| GCNII-SCT | 0.017 | 0.018 | 0.017 | 0.020 | 0.025 |

Table 9: Average computational time per epoch for five heterophilic datasets with fixed $48/32/20\%$ splits. The depth of each model is 8 layers with 16 hidden channels. (Unit: second)

