# OpenReview forum: "Learning to Control the Smoothness of GCN Features"
_NeurIPS.cc/2024/Conference — Submitted to NeurIPS 2024_

### Official Review · Reviewer_RBEx · 2024-07-08

**Soundness:** 3
**Presentation:** 2
**Contribution:** 3
**Rating:** 5
**Confidence:** 3

**Summary:**

The paper "Learning to Control the Smoothness of GCN Features" investigates the impact of activation functions, specifically ReLU and leaky ReLU, on the smoothness of node features in Graph Convolutional Networks (GCNs). It provides a geometric characterization of these effects, showing how altering the input's projection onto eigenspace M can control the smoothness of output features. The study introduces a Smoothness Control Term (SCT) to modulate the smoothness of node features, aiming to improve node classification tasks in both homophilic and heterophilic graphs. Experimental results validate the efficacy of SCT, demonstrating significant improvements in node classification accuracy for several GCN-style models .

**Strengths:**

## Originality
- The paper introduces a novel approach to control the smoothness of Graph Convolutional Networks (GCNs) features, which is a significant departure from traditional methods. It builds upon and extends the work of Oono & Suzuki and Cai & Wang by integrating geometric insights with the message-passing process in GCNs.

## Quality
- The paper provides a robust theoretical framework, including geometric characterizations and proofs, that underpin the proposed methods.
- Extensive experiments validate the theoretical claims, showing significant improvements in node classification accuracy. Detailed descriptions of the experimental setup, including datasets and hyperparameter tuning, enhance the reproducibility of the results.

## Clarity
- The paper is well-structured, with clear sections that logically flow from introduction to theoretical analysis, experimental validation, and conclusions.
- The paper provides a comprehensive review of related work, situating its contributions within the broader context of graph neural networks research.

## Significance
- The ability to control the smoothness of GCN features addresses a to me interesting and important challenge in graph neural networks, with potential applications in various domains such as social network analysis, biological networks, and recommendation systems.
- The proposed SCT shows improvements in real-world datasets.
- The insights gained from this work could inform future research on activation functions and feature smoothness in other types of neural networks.

**Weaknesses:**

## Weaknesses
- The removal of white space in the paper makes it hard to read. The authors should really try and make the paper easier to read visually by not condensing as much math in the main text as possible. Not only is this arguably in violation of the guidelines, but it also illustrates that the authors need to distinguish clearer what the main contributions are and which parts in the main text can go to an appendix.
- While the geometric insights provided are valuable, the complexity of the mathematical formulations is challenging for readers not well-versed in advanced geometry and spectral graph theory. Simplifying explanations or providing more intuitive examples could enhance accessibility. Moreover, I feel like the math could be made more intuitive by giving verbal explanations before the theorems. The cramming of the paper and not highlighting enough what the main contributions should be improved.
- Although the paper compares SCT with a few baseline models, it would benefit from a broader comparison with additional state-of-the-art methods in GCNs and GNNs to provide a more comprehensive evaluation of its effectiveness.
- The experiments are primarily conducted on benchmark datasets. Incorporating more real-world applications and diverse datasets would demonstrate the practical relevance and versatility of the proposed method.
- The drop in accuracy for deeper models (16 or 32 layers) is noted but not deeply analyzed. A more thorough investigation into the causes of this performance degradation, beyond mentioning vanishing gradients, could offer insights into potential improvements. What happens if you use techniques that combat eg oversmoothing?
- While the paper mentions computational efficiency, a more detailed discussion on the computational overhead introduced by SCT, including potential trade-offs between accuracy and efficiency, would be beneficial.

**Questions:**

## Questions

-  Could you provide more intuitive examples or visual aids to help readers better understand the geometric characterization of smoothness in GCN features?
- Why did you choose the specific baselines for comparison? How would SCT perform against other state-of-the-art GCN and GNN models not included in your study?
- Can you elaborate on the causes of performance degradation in deeper models (16 or 32 layers)? Have you considered any specific techniques to mitigate this issue?

**Limitations:**

## Limitations

The authors have acknowledged several limitations of their work, including the over-smoothing issue in deep GCNs and the dependence on specific activation functions, but they could improve by providing more detailed discussions on computational efficiency and potential negative societal impacts.

---

> ### Author Rebuttal · Authors · 2024-08-05
>
> Thank you for your thoughtful review and valuable feedback. In what follows, we provide point-by-point responses to your comments on the weaknesses and limitations of our paper and your questions.
>
> ---
>
> **1: The removal of white space in the paper makes it hard to read.**
>
> **Response:** We appreciate your feedback and we are happy to move certain parts to the appendix to make the paper easier to read.
>
> ---
>
> **2: The math could be made more intuitive by giving verbal explanations before the theorems. The cramming of the paper and not highlighting enough what the main contributions should be improved. Could you provide more intuitive examples or visual aids to help readers better understand the geometric characterization of smoothness in GCN features?**
>
> **Response:** Again, we appreciate your comment. We have summarized the main contributions in verbal explanations in Section 1.1. We also provide navigation for each contribution to the particular sections for details. In Lines 110-122, we provide a very brief review of used results from spectral graph theory.
>
> Regarding the geometric insight, all we established is that there is a high-dimensional sphere associated with the input and output of ReLU and leakyReLU. In Sections 3.1 and 3.2, we provide details on the center and radius of these spheres.
>
> ---
>
> **3:  Why did you choose the specific baselines for comparison? How would SCT perform against other SOTA GCN and GNN models not included in your study? It would benefit from a broader comparison with additional SOTA methods in GCNs and GNNs to provide a more comprehensive evaluation of its effectiveness.**
>
> **Response:** We design the experiments to solidify our theoretical results and verify the effectiveness of the informed algorithm. In particular, we have the following two purposes in mind: First, SCT can avoid over-smoothing for GCN. Second, learning to balance the smooth and non-smooth features can improve the performance of GCN-style models that even do not suffer from over-smoothing. We choose two state-of-the-art GCN-style models - GCNII and EGNN - as the testbeds. To show the effectiveness of the proposed approach, we have comprehensively studied models with different layers on 10 different benchmark datasets, covering all datasets from the papers we have benchmarked on.
>
> Our proposed SCT can be treated as a bias term for GCN-style models and it can be integrated with many other GNNs. Studying other GNNs can be interesting for future work, especially whether the proposed SCT can improve the performance of other GNNs. We are happy to provide some comments in the revised paper.
>
> ---
>
> **4: Incorporating more real-world applications and diverse datasets would demonstrate the practical relevance and versatility of the proposed method.**
>
> **Response:** Our work is primarily theoretical with an informed practical algorithm. Using diverse existing celebrated benchmark tasks is a crucial step in evaluating the work. To better contrast with existing works, we used tasks from existing papers. We have further tested the performance of SCT on the peptide dataset originating from biophysics applications; our results in Table 6 of the rebuttal file further confirm the effectiveness and versatility of the proposed SCT.
>
> ---
>
> **5: The drop in accuracy for deeper models is noted but not deeply analyzed. A more thorough investigation into the causes of this performance degradation, beyond mentioning vanishing gradients, could offer insights into potential improvements. What happens if you use techniques that combat eg oversmoothing? Can you elaborate on the causes of performance degradation in deeper models? Have you considered any specific techniques to mitigate this issue?**
>
> **Response:** The drop in accuracy for deeper models occurs for GCN and GCN-SCT but not for other models that have skip connections in their architectures, which motivates us to investigate the vanishing gradient issue. It is noted in the original ResNet paper that using skip connections can effectively alleviate vanishing gradients in training deep networks. Our results in Figure 4 in the appendix confirm that vanishing gradients occur for GCN and GCN-SCT but not for other models.
>
> The skip connection has become a celebrated technique to mitigate the vanishing gradient issue, and this is also the case for training deep GCNs.
>
> ---
>
> **6: While the paper mentions computational efficiency, a more detailed discussion on the computational overhead introduced by SCT, including potential trade-offs between accuracy and efficiency, would be beneficial.**
>
> **Response:** The computational overhead introduced by SCT is not very significant compared to the whole cost of training and deploying GNN-style models. Notice that we can avoid performing eigendecomposition by using the fact that the basis of the space $\mathcal{M}$ -- eigenspace associated with the largest eigenvalue of the message-passing matrix -- is given by the indicator functions of each connected component of the graph; see Lines 110-114. Therefore, the problem reduces to finding connected components of the graph, and an efficient approach to identifying connected components for undirected graphs is using disjoint set union (DSU). Initially declare all the nodes as individual subsets and then visit them. When a new unvisited node is encountered, unite it with the under. In this manner, a single component will be visited in each traversal. The time complexity is $O(V)$.
>
> We provide the computational time for models with and without SCT for Ogbn-arxiv in Table 2 of the rebuttal file.
>
> ---
>
> **7: Providing more detailed discussions on computational efficiency and potential negative societal impacts.**
>
> **Response:** We will include these in the revision. See our response the point 6 above for computational efficiency.
>
> ---
>
> Thank you for considering our rebuttal. We appreciate your feedback and are happy to address further questions on our paper.

---

> > ### Comment · Reviewer_RBEx · 2024-08-12
> >
> > Thanks a lot for the rebuttal. I will raise my score to a 5, but really urge the authors to reformat the paper in a better readable way.

---

> > > ### Author Response · Authors · 2024-08-12
> > > **Thank you**
> > >
> > > Thank you for considering our rebuttal and support of our work. We will reformat the paper following your and other reviewers' suggestions.

---

### Official Review · Reviewer_5MTH · 2024-07-09

**Soundness:** 2
**Presentation:** 2
**Contribution:** 2
**Rating:** 5
**Confidence:** 4

**Summary:**

The paper addresses the challenge of balancing smooth and non-smooth features in graph convolutional networks (GCNs) for node classification. Building on previous work that highlighted the correlation between feature smoothness and classification accuracy, the authors propose a novel method to control the smoothness of node features through a geometric approach and an augmented message-passing process. Their strategy involves establishing a geometric relationship between input and output vectors of activation functions like ReLU and Leaky-ReLU, and integrating a learnable term in the graph convolutional layers to modulate feature smoothness. The paper provides an empirical study to showcase the effectiveness of the proposed method.

**Strengths:**

The paper offers a novel geometric insight into the effects of different activation functions, specifically ReLU and Leaky-ReLU, on smoothing in GCNs. Furthermore, it introduces an innovative method to enhance the message-passing framework for GCNs (and similar networks) to better control smoothing.

**Weaknesses:**

* The empirical results section is very weak and a major revision is needed in order to be able to judge the merit of the proposed method. The most significant weaknesses are:
    * The paper does not compare to established baselines from other competing methods. For instance, one could take a look at the tables in https://arxiv.org/abs/2202.04579 or https://arxiv.org/abs/2210.00513 and add the results to the comparison in this paper (i.e., for cornell, texas, wisconsin, squirrel, and so on).
    * The paper reports different (mostly weaker) performance for the baseline models such as GCN and GCNII. Again, this should be fixed and established results can be taken from the tables in the two papers mentioned above.
    * The improvement using the proposed SCT is very marginal in many experiments, and in particular the performance still drops (or improves very little) for higher number of layers. It is somewhat expected that increasing the number of layers and solving oversmoothing does not help much in homophilic tasks, i.e., the tasks considered in Table 1. However, if the method works it should lead to significant improvements on heterophilic tasks, even for increasing number of layers. Therefore, please show the results for the datasets in Table 2 in the same way as you show them in Table 1, i.e., for increasing number of layers.
    * It is claimed in Table 1 that the drop in performance of GCN for increasing number of layers is due to vanishing gradients and not due to oversmoothing. This claim has to be justified empirically.
    * Figure 4 in the appendix showing gradient norms is not meaningful. The y-axis should be displayed on a logarithmic scale. Vanishing gradients occur for gradients approaching zero (exponentially fast). It is not clear if this is the case here, since it is plotted in linear scale.

* The structure and readability of this paper should be improved. For instance, it would be helpful for readers who are not very familiar with the graph-learning field, to introduce the concept of GNNs and GCNs in the introduction. It would be advisable to move the technical aspects from section 1 to section 2. In fact, none of the definitions presented at the beginning of the introduction are needed anywhere else in the remainder of the introduction section. Thus, this could be moved to section 2.

**Questions:**

how expensive is it to compute the eigenbasis for the smoothness modulation term (5)? How does it scale with respect to the number of nodes and edges? How long does it take to compute for the biggest graph considered here, i.e., arxiv graph? Is that a limitation for large-scale graph learning?

**Limitations:**

* The method necessitates a pre-processing step to compute the eigenbasis in equation (5). This may not scale efficiently for very large graphs.
* The method does not show significant improvements. Notably, there are more effective models and methods available for mitigating oversmoothing, which have not been cited or empirically compared. Examples include https://arxiv.org/abs/2202.04579, https://arxiv.org/abs/2210.00513, https://arxiv.org/abs/2206.05437, https://arxiv.org/abs/2110.14446, https://arxiv.org/abs/2206.10991,
https://arxiv.org/abs/2006.11468.
* While the theoretical justification of the approach is intriguing, the insights are somewhat limited. For example, there are no provided estimates for choosing the parameter alpha, which instead has to be learned through gradient descent.
* The empirical results are not convincing.

---

> ### Author Rebuttal · Authors · 2024-08-06
>
> Thank you for your thoughtful review and valuable feedback. In what follows, we provide point-by-point responses to your comments on the weaknesses and limitations of our paper and your questions.
>
> ---
>
> **1: The paper does not compare to established baselines from other competing methods. E.g. tables in arxiv:2202.04579 or arxiv:2210.00513 and add the results to the comparison in this paper.**
>
> **Response:** Thank you for pointing out these papers to us.
>
> Our choice of architecture is based on the baseline of EGNN and we incorporate architectures in a comparable manner using the PyTorch Geometric framework. We provide full baseline architectures and hyperparameter details for all experiments. This enables us to consistently study the layer-wise bias introduced by SCT. The reviewer recommended papers use different architectures -- each model has a customized architecture, making it difficult to study the effects of SCT (a layer-wise bias term) introduced in different models.
>
> Of the reviewer-recommended works, the model that makes the most sense for comparison to our method is G2-GCN. Notice that they do not provide details regarding the optimal hyperparameters selected for each experiment in either the paper or source code. Instead, they provide a range of randomly chosen parameters in the paper. It is beyond the purview of this work to fine-tune G2-GCN and so a reasonable hyperparameter selection was chosen. We compare the G2-GCN architecture with and without SCT. The results are listed in Table 4 of the rebuttal pdf.
>
> As pointed out by Reviewer pS2J, our numerical results show that the proposed method is effective for various data sets and models, showing the versatility of the proposed method.
>
> ---
>
> **2: The paper reports different (mostly weaker) performance for the baseline models such as GCN and GCNII.**
>
> **Response:** The GCN and GCNII architectures we used are different from those in the reviewer-mentioned papers. Our used architectures provide a direct comparison between layer-wise operations performed by the addition of the SCT bias. We do not claim to achieve state-of-the-art performance but provide a reasonable baseline to compare methodologies. This provides a testbed for verifying our theoretical results. We utilize standard implementations of the GCN and GCNII layers using PyTorch Geometric. For additional comparison, Table 3 of the rebuttal pdf provides results for including SCT in the GCN architectures mentioned by the reviewer. Again, SCT improves the baseline model with a noticeable margin.
>
>
> ---
>
> **3: Please show the results for the datasets in Table 2 in the same way as you show them in Table 1, i.e., for increasing number of layers.**
>
> **Response:** In Table 5 of the rebuttal PDF file, we provide the corresponding table for the Texas dataset. We provide results for only the Texas dataset due to the space limitation, similar results hold for other datasets based on our experiments.
>
> ---
>
> **4: Justify the claim that the accuracy drop in Table 1 is due to vanishing gradients and not oversmoothing.**
>
> **Response:** We investigate the vanishing gradient issue in training deep GCN and GCN-SCT by plotting the gradient norm in Fig. 4 in the appendix, confirming that the vanishing gradient occurs for GCN and GCN-SCT but not for other models.
>
> ---
>
> **5: Figure 4 in the appendix showing gradient norms is not meaningful. The y-axis should be displayed on a logarithmic scale.**
>
> **Response:** We appreciate your feedback. In the rebuttal file, we have provided the updated figures in Figure 1. We also show the exponential decay in gradient norms.
>
> ---
>
> **6: The structure and readability of this paper should be improved.**
>
> **Response:** We are happy to incorporate your comments in the revision.
>
> ---
>
> **7: How expensive is it to compute the eigenbasis for the term (5)? How does it scale w.r.t. the number of nodes and edges? How long does it take to compute for the biggest graph considered here, i.e., arxiv graph?**
>
> **Response:** The computational overhead introduced by SCT is not very significant compared to the entire cost of training and deploying GNNs. Notice that we can avoid performing eigendecomposition by using the fact that the basis of the space $\mathcal{M}$ is given by the indicator functions of each connected component of the graph; see Lines 110-114. Therefore, the problem reduces to finding connected components of the graph, and an efficient approach to identifying connected components for undirected graphs is using disjoint set union (DSU). Initially declare all the nodes as individual subsets and then visit them. When a new unvisited node is encountered, unite it with the under. In this manner, a single component will be visited in each traversal. The time complexity is $O(V)$.
>
> We provide the computational time for models with and without SCT for Ogbn-arxiv in Table 2 in the rebuttal file.
>
> ---
>
> **8: Significance of the results. There are more effective models and methods available for mitigating oversmoothing, which have not been cited or empirically compared.**
>
> **Response:** Please refer to our response to your first comment on the significance of numerical results. We appreciate the reviewer pointing out these references to us; we are happy to cite them in the revision.
>
> ---
>
> **9: There are no provided estimates for choosing the parameter alpha, which instead has to be learned through gradient descent.**
>
> **Response:** The desired smoothness that favors node classification is task-dependent and unknown. As such, we make the model learn to automatically balance smooth and non-smooth features; our empirical results show that such a learning-based strategy does improve the performance of some remarkable GCN-style models.
>
> ---
>
> Thank you for considering our rebuttal. We appreciate your feedback and are happy to address further questions on our paper.

---

> > ### Comment · Reviewer_5MTH · 2024-08-13
> >
> > Unfortunately, I cannot see any of the changes the authors claim to have made.

---

> ### Author Response · Authors · 2024-08-13
> **Further clarification**
>
> Dear Reviewer 5MTH,
>
>    Thank you for your response. Additional experimental results are provided in the one-page pdf file and the other changes have been made in the revised paper, which is straightforward to do. We cannot update the paper on openreview per the rebuttal rule. The system does not allow us to upload the revised version.
>
>
> Regards,
>
> Authors

---

> > ### Author Response · Authors · 2024-08-13
> > **Further clarification -- cont'd**
> >
> > Dear Reviewer 5MTH,
> >
> > We first appreciate your further feedback. We are not sure if we misunderstood your comment that ``unfortunately, I cannot see any of the changes the authors claim to have made.’’ We are happy to address your further comments before the end of the discussion period.
> >
> > We believe we have provided detailed responses with additional experimental results in the rebuttal file - see the general response. We have also restructured the paper following the reviewers' feedback and added the additional results together with the references pointed out by the reviewer to the revised paper. Again, the openreview does not allow us to update the paper. Moreover, the rebuttal instruction says we can only post a single-page PDF file for additional figures and tables.
> >
> >
> >
> > Regards,
> >
> > Authors

---

> > > ### Comment · Reviewer_5MTH · 2024-08-13
> > >
> > > Thank you for pointing this out. Indeed, I was only looking at the revised paper, not the rebuttal pdf.
> > > I appreciate the depth and detail of the author's response, which has clarified my concerns. I will increase my score accordingly. That being said, it is very important that the authors include the computational time comparison on the arxiv dataset in the paper as well as a discussion on it, as the computational burden of computing the eigenbasis was pointed out by almost all reviewers.

---

> > > > ### Author Response · Authors · 2024-08-13
> > > > **Thank you**
> > > >
> > > > Thank you for your further feedback and support of our paper. Your thoughtful review and valuable comments have significantly improved our paper.

---

### Official Review · Reviewer_DP5C · 2024-07-10

**Soundness:** 3
**Presentation:** 2
**Contribution:** 3
**Rating:** 5
**Confidence:** 4

**Summary:**

The paper studies how GCN smoothes node features in terms of unnormalized and normalized smoothness. The results show that adjusting projection can alter the normalized smoothness to any desired level. Based on this, the paper proposes a new method SCT to let GCN learn node features with a desired smoothness to enhance node classification and verifies it effectiveness in practice.

**Strengths:**

1.	Understanding the effect of nonlinearities in GNNs is an important yet underexplored problem due to the complexity of nonlinearities. The paper offers a new perspective on it.

2.	Oversmoothing is a known issue, while it is also known that some amount of smoothness is desired for graph learning. How to find the ideal amount of smoothness among node features is an important but nontrivial problem.

3.	The proposed method SCT is principled and seems effective.

**Weaknesses:**

1.	While I can see that normalized smoothness has its own merit, this notion could be better motivated and connected to the literature.

   a.	Given analysis in [27, 4] is asymptotic, I would not say that “over-smoothing – characterized by the distance of features to eigenspace M or the Dirichlet energy – is a misnomer”, as it is too strong of a claim to make. Those results essentially say that “very deep GCNs are bad due to oversmoothing” which has their limitations but can be well justified by the distance of features to eigenspace M or the Dirichlet energy.

   b.	My understanding is that normalized smoothness could be more connected to the non-asymptotic notion of oversmoothing studied in [32], which is defined based on the Bayes error of classification (the distance to the decision boundary) and hence taken the magnitude of features into account.

   c.	Based on the above, the argument presented in the paper can be strengthened in the following way: the motivation of normalized smoothness should be based on a discussion on how the magnitude (and hence normalized smoothness) is more related to a non-asymptotic notion of oversmoothing, which is directly related to the classification performance of finite-depth/shallow GNNs. Based on this, the results present in this paper is more practically relevant than the previous asymptotic result.

I would suggest the authors modify the relevant text in the introduction and analysis accordingly.


2.	The analysis only applies for GCNs, while whether the analysis or the proposed method can be extended to more complexed GNNs such as GATs or graph transformers is unclear.

3.	The analysis only applies for ReLU and LeakyReLU, which reads a bit specific. I wonder if the results can be generalized to a general family of nonlinearities.

4.  For the experiments, there is a lack of baseline comparisons except the basic backbone architecture. For example, I wonder how it would compare to APPNP, which is proposed to balance the need to explore larger neighborhood and locality and the implicit goal is also to produce node features with the "right" amount of smoothness.

4.	Another presentation suggestion I have for the authors is that one should minimize the use of in-text math and bold or italic fonts for highlighting (such as line 167-173).  Math is hard to read in-text and when too many texts are highlighted, the paper becomes ever harder to read because everything seems to be emphasized and it kind of messes up with its original purpose.

**Questions:**

See weaknesses.

---

> ### Author Rebuttal · Authors · 2024-08-05
>
> Thank you for your thoughtful review, valuable feedback, and endorsement. We appreciate your invaluable suggestions and will revise the paper accordingly. In what follows, we provide point-by-point responses to your comments on the weaknesses of the paper.
>
> ------
>
> **1. While I can see that normalized smoothness has its own merit, this notion could be better motivated and connected to the literature.**
>
> **Response:** We appreciate all your invaluable suggestions and are happy to incorporate these suggestions in the revision.
>
> The normalized smoothness notion was pointed out in [4], and we adapted this notion and noticed the equivalence between Dirichlet energy and the orthogonal complement of the projection of features onto the eigenspace. The analysis in [27,4] is asymptotic, but both normalized or unnormalized smoothness notions can characterize the smoothness of node features for GCN with a finite number of layers. A particular motivation for our work is the empirical correlation between the accuracy of GCN and a normalized smoothness-related quantity studied in [27]. As such, we propose a practical approach to control the smoothness of GCN features based on our observed geometric relationship between the input and output of activation functions.
>
> ------
>
>
> **2: The analysis only applies for GCNs, while whether the analysis or the proposed method can be extended to more complexed GNNs such as GATs or graph transformers is unclear.**
>
> **Response:** It has recently been proved in [A] that attention-based GNNs also suffer from over-smoothing. Specifically, [A] demonstrates that the node features converge to the same eigenspace as those in GCN. While using a different smoothness notion, they show that their measurement is equivalent to the one used by Oono & Suzuki (ICLR, 2020) and our work. Consequently, since the characterization of smoothness is the same, our technique can be applied to these frameworks to control normalized smoothness. In particular, in our analysis, $\bf z$ can be the feature obtained through any graph convolution, including attention-based methods. We will incorporate these discussions in the revision.
>
> The main reason we did not include these models in our work is that, within the scope of "controlling smoothness" (which is not the same as mitigating over-smoothing), to the best of our knowledge, we only found EGNN to be an example in this branch. This is why we included it in our comparisons. Moreover, since it is built upon GCN and GCNII, we also included these two models.
>
> [A] Wu, Xinyi, et al. "Demystifying oversmoothing in attention-based graph neural networks." NeurIPS, 2023.
>
>
> ------
>
>
> **3: The analysis only applies for ReLU and LeakyReLU, which reads a bit specific. I wonder if the results can be generalized to a general family of nonlinearities.**
>
> **Response:** The geometric characterization of the effects of activation functions in Section 3 uses some particular piecewise linear properties of ReLU and LeakyReLU. We expect a similar analysis can be applied to some other piecewise linear activation functions.
>
> Some analyses of the achievable smoothness using the proposed SCT (in Section 4) can be extended to more general activation functions like ELU and SELU; see our discussion in Lines 233-235.
>
> ------
>
>
> **4: For the experiments, there is a lack of baseline comparisons except the basic backbone architecture. For example, I wonder how it would compare to APPNP, which is proposed to balance the need to explore larger neighborhood and locality and the implicit goal is also to produce node features with the "right" amount of smoothness.**
>
> **Response:** We have incorporated APPNP into our experimental results. We train APPNP using the optimal hyperparameters as reported by the APPNP paper. The results are reported in Table 4 of the rebuttal file.
>
>
> ------
>
>
> **5: Another presentation suggestion I have for the authors is that one should minimize the use of in-text math and bold or italic fonts for highlighting (such as line 167-173).**
>
> **Response:** We appreciate your suggestion and will account for this in the revision.
>
>
> ------
>
>
> Thank you for considering our rebuttal. We appreciate your feedback and are happy to address further questions on our paper.

---

> > ### Comment · Reviewer_DP5C · 2024-08-12
> > **Thank you**
> >
> > I thank the authors for responding to my review. One difficulty I can see regarding extending to GATs or GTs is that unlike graph convolutions considered in this paper, attention matrices are in general not symmetric and the current analysis techniques developed for symmetric matrices might not be trivially applied. Nonetheless, the rebuttal has addressed the rest of my concerns. I will keep my score for now and stay on the positive side.

---

> > > ### Author Response · Authors · 2024-08-12
> > > **Thank you for considering our rebuttal**
> > >
> > > Thank you for your further feedback and your support of our work.
> > >
> > > We would like to clarify that while graph attention networks (GATs) may employ asymmetric graph convolutions, our work relies solely on the characterization of oversmoothing. As demonstrated in [1], GATs with asymmetric attention matrices can still exhibit oversmoothing, and our oversmoothing measurement is equivalent to that used in [1]. Importantly, our proposed method, smoothness control term (SCT), analyzes the impact of activation functions on the smoothness of $\bf Z = \bf W\bf H\bf G$, by considering only the input $\bf Z$ and output $\sigma(\bf Z)$ of these functions. This abstraction allows us to isolate the effects of the weight matrices $\bf W$ and graph convolution $\bf G$, providing an understanding of how SCT controls the smoothness of $\sigma(\bf Z)$.
> > >
> > >
> > >
> > > [1] Wu, Xinyi, et al. "Demystifying oversmoothing in attention-based graph neural networks." NeurIPS, 2023.
> > >
> > >
> > > Thank you again for considering our rebuttal.

---

### Official Review · Reviewer_pS2J · 2024-07-10

**Soundness:** 3
**Presentation:** 2
**Contribution:** 3
**Rating:** 6
**Confidence:** 3

**Summary:**

This paper first shows that in GCN, the output of ReLU or LeakyReLU lies on a sphere whose input is characterized by components parallel and perpendicular to $\mathcal{M}$, the space spanned by eigenvectors for the maximum eigenvalue of a graph. As a corollary, this paper shows that these activation functions do not increase the component of the feature vector perpendicular to $\mathcal{M}$. Furthermore, this paper defines the normalized smoothness and evaluates how its range varies with the activation functions. Based on this discussion, this paper proposes an SCT that learns the parallel components of the feature vectors. The proposed method is applied to GCN, GCNII, and EGNN, and verifies its effectiveness by applying it to node prediction tasks with various heterophily.

**Strengths:**

- The theorems presented in the theoretical analysis (Propositions 3.2 and 3.3) enable a unified treatment of ReLU and Leaky ReLU.
- Numerical experiments show that the proposed method is effective for various data sets and models, showing the versatility of the proposed method.

**Weaknesses:**

- I need help understanding the explanation in Section 3.3. More specifically, it is difficult to understand that the *independence* of the inequality means that the upper bound of the inequality does not depend on the value of $\boldsymbol{Z}_{\mathcal{M}}$. I suggest writing it explicitly.
- P7, L.265: It seems strange that although SCT changes its architecture depending on whether the underlying GNN is GCT or GCNII, it has the same name. I suggest naming SCT for GCT and SCT for GCNII differently.

**Questions:**

P5, L.196: $\boldsymbol{e}$ -> $\boldsymbol{e}_{1}$
P6, L.209: What is the value of $a$ of $\sigma_a$?

**Limitations:**

The authors discuss in the conclusion section that the proposed method's limitation is that it assumes the model's oversmoothing. However, I do not think this is a limitation because if the model does not cause oversmoothing, there is no need to use the proposed method. Rather, I recommend evaluating whether SCT has bad effects when the model does not cause oversmoothing.

---

> ### Author Rebuttal · Authors · 2024-08-01
>
> Thank you for your thoughtful review, valuable feedback, and endorsement. In what follows, we provide point-by-point responses to your comments on the weaknesses and limitations of our paper and your questions.
>
>
> ------
>
> **1: I need help understanding the explanation in Section 3.3. More specifically, it is difficult to understand that the independence of the inequality means that the upper bound of the inequality does not depend on the value of $\mathbf{Z}_{\mathcal{M}}$. I suggest writing it explicitly.**
>
> **Response:**
> To understand this better, we consider two inputs ${\bf Z}, {\bf Z}’\in \mathbb{R}^{d\times n}$, where ${\bf Z}\_{\mathcal{M}^\perp}={\bf Z}’\_{\mathcal{M}^\perp}$ but ${\bf Z}\_{\mathcal{M}} \neq {\bf Z}’\_{\mathcal{M}}$. Moreover, let ${\bf H}=\sigma(\bf Z)$ and $\bf H’=\sigma(\bf Z’)$ with $\sigma$ being ReLU or leaky ReLU. Based on our geometric results in Sections 3.1 and 3.2, we have $||\bf H||\_{\mathcal{M}^\perp}\leq ||\bf Z||\_{\mathcal{M}^\perp}$ and $||\bf H’||\_{\mathcal{M}^\perp}\leq ||\bf Z’||\_{\mathcal{M}^\perp}$. Notice that $\bf Z\_{\mathcal{M}^\perp}=\bf Z’\_{\mathcal{M}^\perp}$, we have $||\bf H’||\_{\mathcal{M}^\perp}\leq ||\bf Z||\_{\mathcal{M}^\perp}$.
>
> Notice that $||\bf H’||\_{\mathcal{M}^\perp}\leq ||\bf Z||\_{\mathcal{M}^\perp}$ indicates that altering $\bf Z$ to any $\bf Z’$ with $\bf Z\_{\mathcal{M}^\perp}$ being preserved does not change the fact that $||\bf H’||\_{\mathcal{M}^\perp}$ will be upper bounded by $ ||\bf Z||\_{\mathcal{M}^\perp}$.
>
> We have revised our paper per your suggestion.
>
>
> ------
>
> **2: P7, L.265: It seems strange that although SCT changes its architecture depending on whether the underlying GNN is GCT or GCNII, it has the same name. I suggest naming SCT for GCT and SCT for GCNII differently.**
>
> **Response:** Thank you for your suggestion, we will name them to specify the use of different SCT architectures in the revised paper.
>
> ------
>
> **3: P5, L.196:  ${\bf e}\rightarrow {\bf e}\_1$  P6, L.209: What is the value of $\alpha$ of $\sigma_{\alpha}$?**
>
> **Response:**
> L196: For the sake of notation, we denote ${\bf e}_1$ as ${\bf e}$ as pointed out in L193-194. We are happy to denote it as ${\bf e}_1$.
>
> L209: $\alpha$ is a smoothness control parameter and we vary this parameter from -1.5 to 1.5.
>
>
> ------
>
>
> **4. I recommend evaluating whether SCT has bad effects when the model does not cause oversmoothing.**
>
> **Response:** Thank you for your suggestion. Indeed, we have evaluated the effect of SCT on GCNII and EGNN - two state-of-the-art GCN-style models that do not suffer from over-smoothing. Our numerical results show that SCT can further improve the performance of these models. Though GCNII and EGNN do not suffer from over-smoothing, the node features learned by these two models do not necessarily have a good balance between smooth and non-smooth components that are optimal for node classification. In fact, we can improve the performance of these models by letting the model automatically learn a good balance between smooth and non-smooth features.
>
> On the one hand, it is crucial to avoid over-smoothing for node classification. On the other hand, as noticed empirically by Oono & Suzuki (ICLR, 2020) the ratio between smooth and non-smooth features is highly correlated with the classification accuracy. Based on this empirical observation and our established geometric insights, we have proposed SCT to let GCN-style models automatically learn features with a desired smoothness to improve node classification.
>
>
> ------
>
>
> Thank you for considering our rebuttal. We appreciate your feedback and are happy to address further questions on our paper.

---

> > ### Comment · Reviewer_pS2J · 2024-08-12
> >
> > I thank the authors for responding to my review comments. Their responses answered my questions appropriately. So, I want to keep my scores.

---

> > > ### Author Response · Authors · 2024-08-12
> > > **Thank you**
> > >
> > > Thank you for considering our rebuttal.

---

### Official Review · Reviewer_6k42 · 2024-07-11

**Soundness:** 3
**Presentation:** 2
**Contribution:** 3
**Rating:** 4
**Confidence:** 4

**Summary:**

The paper deals with (Over-)smoothing is Graph Neural Networks. While it was previously known that GCN-type GNNs oversmooth, this paper reexamines the case for GCN-type architectures with Relu-type activation functions in terms of *normalized* smoothness. The authors show that the convergence behaviour of GCNs can be split into two parts, a "smooth" part and a "non-smooth" part and that by manipulating the smooth part, one can influence the normalized smoothness of the signal. From these theoretical insights, the authors propose a new system that uses a learnable parameter that modulates the smooth part, making the model able to learn the most beneficial normalized smoothness for the problem at hand.

**Strengths:**

- The paper furthers the understanding of oversmoothing in GNNs
- The proposed approach is well-founded in theory

**Weaknesses:**

- The experiments are unconvincing.
    - There is a slight improvement to be found in models with SCT, but this is not too surprising, as these models also have more parameters, and mostly improvements are quite slim.
    - You choose GCN, GCNII and EGNN, two of which have built-in skip connections, that are known to help with oversmoothing. This also coincides with the models that cope quite well with a larger number of layers. The vanilla GCN takes a huge hit in all benchmarks apart from ogbn-arxiv. So learning the normalized smoothness of features does not actually seem to help in the case of GCN.
    - The other two models don't suffer from oversmoothing to begin with

**Questions:**

- The parameter $\alpha$ can be chosen such that the normalized smoothness does not diminish asymptotically. However, the unnormalized smoothness does; does this mean that the norm of the features $||z|| \rightarrow 0$ tends towards 0? Doesn't this mean that features are also unusable in deeper layers?

- Is this analysis extendable to other architectures? E.g. GAT and other attention-based methods are also known to oversmooth, can a similar trick be applied there?

- Table 5 details the hyperparameters that were tried for each model. These seem to be very many combinations. Is it correct that you tried approx. 1.2 Million hyperparameter combinations for EGNN?

**Limitations:**

The limitations are not well-discussed.
The only limitation the authors claim for this work is, that: "without this condition [that oversmoothing happens], SCT cannot ensure performance guarantees." There are no performance guarantees given for SCT. This is the only limitation discussed.

---

> ### Author Rebuttal · Authors · 2024-08-05
>
> Thank you for your thoughtful review and valuable feedback. In what follows, we provide point-by-point responses to your comments.
>
> ------
>
> **1. There is a slight improvement to be found in models with SCT, but this is not too surprising, as these models also have more parameters, and mostly improvements are quite slim.**
>
> **Response:** The accuracy improvement is task-dependent, and it is quite substantial for more challenging heterophilic datasets. As shown in Table 2 of our paper, the accuracy improvements are often more than 10%. In the rebuttal file, we have conducted hypothesis testing on the significance of the accuracy improvement due to the proposed SCT. The results in Table 1 of the rebuttal file confirm the statistical significance of accuracy improvement.
>
> The accuracy improvement using SCT is not because of using more parameters: 1) Table 1 shows the baseline model with SCT and 2 layers can often achieve better accuracy than the baseline model without SCT but has 4, 16, or 32 layers. This is especially true for larger graphs like Coauthor-Physics and Ogbn-arxiv. 2) Table 2 shows that the baseline model with SCT and a shallow architecture can significantly outperform the same model without SCT but with a deeper architecture.
>
>
> ------
>
> **2. Learning the normalized smoothness of features does not actually seem to help in the case of GCN.**
>
> **Response:** The accuracy drop for GCN and GCN-SCT with a large number of layers is because of the vanishing gradient issue rather than over-smoothing. We investigate the vanishing gradient issue in training deep GCN and GCN-SCT by plotting the gradient norm in Figure 4 in the appendix.
>
> The drop in accuracy for deeper models does not occur for other models that have skip connections in their architectures. It is noted in the original ResNet paper that using skip connections can effectively alleviate vanishing gradients in training deep networks. Controlling the smoothness of node features cannot help solve the vanishing gradient issue.
>
> ------
>
> **3: GCNII and EGNN don’t suffer from oversmoothing to begin with.**
>
> **Response:** On the one hand, avoiding over-smoothing for node classification is crucial. On the other hand, as noticed empirically by Oono & Suzuki (ICLR, 2020) the ratio between smooth and non-smooth features is correlated with the classification accuracy. Based on this observation and our established geometric insights, we have proposed SCT to let GCN-style models automatically learn features with a desired smoothness to improve node classification.
>
> GCNII and EGNN are two state-of-the-art GCN-style models. We apply SCT to these models to show that learning to balance smooth and nonsmooth features can improve the performance of GCN-style models. Though GCNII and EGNN do not suffer from over-smoothing, the learned node features do not necessarily have a good balance between smooth and non-smooth components that are optimal for node classification. In fact, we can improve the performance of these models by letting the model automatically learn a good balance between smooth and non-smooth features.
>
>
> ------
>
> **4: Does the norm of the features $||z||\rightarrow 0$ tends towards 0? Doesn't this mean that features are also unusable in deeper layers?**
>
> **Response:** Under GCN dynamics, $||z||\rightarrow 0$ as the number of layers goes to infinity. However, it doesn’t mean that features are unusable in deeper layers. Notice that $||z||\neq 0$ for GCN with any finite layer, and the classification results do not change when multiplying features by a constant.
>
> The feature norm itself does not capture the full spectrum of the smoothness of node features; the ratio between smooth and non-smooth features is an alternative smoothness notion.
>
> ------
>
>
> **5: Is this analysis extendable to other architectures? E.g. GAT and other attention-based methods are also known to oversmooth, can a similar trick be applied there?**
>
> **Response:** It has recently been proved in [1] that attention-based GNNs also suffer from over-smoothing. Specifically, [1] demonstrates that the node features converge to the same eigenspace as those in GCN. While using a different smoothness notion, they show that their measurement is equivalent to the one used by Oono & Suzuki (ICLR, 2020) and our work. Consequently, since the characterization of smoothness is the same, our technique can be applied to these frameworks to control normalized smoothness. In particular, in our analysis, $\bf z$ can be the feature obtained through any graph convolution, including attention-based methods. We will incorporate these discussions in the revision.
>
> The main reason we did not include these models in our work is that, within the scope of "controlling smoothness" (which is not the same as mitigating over-smoothing), to the best of our knowledge, we only found EGNN to be an example in this branch. This is why we included it in our comparisons. Moreover, since it is built upon GCN and GCNII, we also included these two models.
>
> [1] Wu, Xinyi, et al. "Demystifying oversmoothing in attention-based graph neural networks." NeurIPS, 2023.
>
> ------
>
> **6: Table 5 details the hyperparameters that were tried for each model. These seem to be very many combinations. Is it correct that you tried approx. 1.2 Million hyperparameter combinations for EGNN?**
>
> **Response:** Thank you for raising this point about the clarity of Table 5. Notice that in [2] Table 4 the optimal hyperparameters are listed for EGNN. We particularly examine tuning over $c\_{max}$, $\alpha$, and $\theta$. We tune GCN using the upper section of Table 2 and use the same selection of values for GCNII but adjust $\alpha$ and $\beta$. There are only 324 combinations and the runtime is under 60s per test.
>
> [2] https://arxiv.org/pdf/2107.02392
>
> ------
>
> Thank you for considering our rebuttal. We appreciate your feedback and are happy to address further questions on our paper.

---

### Official Review · Reviewer_49R3 · 2024-07-11

**Soundness:** 3
**Presentation:** 3
**Contribution:** 4
**Rating:** 8
**Confidence:** 3

**Summary:**

This paper studies how ReLU and Leaky ReLU affect the smoothness of node features in graph convolution layers. The authors demonstrate that adjusting the input projection onto eigenspace $\mathcal{M}$ of the node feature matrix can achieve any desired normalized smoothness. Additionally, they propose a Smoothness Control Term (SCT) to enhance node classification in Graph Convolutional Networks, validated on both homophilic and heterophilic graphs.

**Strengths:**

1. From a geometric perspective, the authors prove that how ReLU and Leaky ReLU affect the smoothness of node features in graph convolution layers.
2. The experimental results validate the theory proposed by the authors.

**Weaknesses:**

Equation 5 implies that using SCT requires performing eigendecomposition. This paper avoids the high time complexity associated with eigendecomposition, especially when the number of nodes in a graph is very large.

**Questions:**

None

---

> ### Author Rebuttal · Authors · 2024-07-31
>
> We thank the reviewer for the thoughtful review, valuable feedback, and endorsement.
>
>
> Regarding the pointed-out weakness, we can avoid performing eigendecomposition by using the fact that the basis of the space $\mathcal{M}$ -- eigenspace associated with the largest eigenvalue of the message-passing matrix -- is given by the indicator functions of each connected component of the graph; see Lines 110-114. Therefore, the problem reduces to finding connected components of the graph, and an efficient way to identify connected components for undirected graphs is using disjoint set union (DSU). Initially declare all the nodes as individual subsets and then visit them. When a new unvisited node is encountered, unite it with the under. In this manner, a single component will be visited in each traversal. The time complexity is $O(V)$.
>
> ------
>
> Thank you for considering our rebuttal. We appreciate your feedback and are happy to address further questions on our paper.

---

### Author Rebuttal · Authors · 2024-08-06

Dear reviewers,

We appreciate your thoughtful reviews and valuable feedback, which have helped us significantly improve the paper. We thank the reviewers’ praise for the originality, quality, clarity, and significance of our work. We are encouraged that reviewers found our proposed approach is well-founded in theory and further the understanding of over-smoothing in GNNs. Moreover, numerical results show that the proposed method is effective for various datasets and models, showing the versatility of the proposed method. We address some common comments from reviewers in this general response, and we provide additional results in the rebuttal PDF file.

---

**1. The computational overhead of SCT**

**Response:** The overhead introduced by SCT is not very significant compared to the entire cost of training and deploying GNN-style models. Notice that we can avoid performing eigendecomposition by using the fact that the basis of the space $\mathcal{M}$ is given by the indicator functions of each connected component of the graph; see Lines 110-114. Therefore, the problem reduces to finding connected components of the graph, and an efficient way to identify connected components for undirected graphs is using disjoint set union (DSU). Initially, declare all the nodes as individual subsets and then visit them. When a new unvisited node is encountered, unite it with the under. In this manner, a single component will be visited in each traversal. The time complexity is $O(V)$.

---

**2: Significance of the experimental results and some additional results**

**Response:** We design the experiments to solidify our theoretical results and verify the effectiveness of the informed algorithm. In particular, we have the following two purposes in mind: First, the proposed SCT can avoid over-smoothing for GCN. Second, by learning to balance the smooth and non-smooth features, SCT can improve the performance of GCN-style models that even do not suffer from over-smoothing. We choose two state-of-the-art GCN-style models - GCNII and EGNN - as the testbeds. To showcase the effectiveness of the proposed approach, we have comprehensively studied models with different layers on 10 benchmark datasets, including all datasets from papers we have benchmarked on.

In the rebuttal file, we provide a few additional numerical results to further confirm the effectiveness of our proposed approach:

- We perform hypothesis testing in Table 1 to test the significance of accuracy improvement for the cases when accuracy does not have a large increase in absolute value.

- We compare the number of parameters and the computational time of models with and without SCT in Table 2.

- We show SCT can improve the performance of the model mentioned by Reviewer 5MTH in Table 3.

- We further benchmark against APPNP in Table 4.

- We provide more detailed results to complement the results reported in Table 2 of our paper; c.f. Table 5 in the rebuttal file.

- We further conduct experiments on the peptide dataset to diversify application scenarios.; c.f. Table 6.

- We provide results to further verify the vanishing gradient issue in training deep GCN and GCN-SCT; c.f. Figure 1 in the rebuttal PDF file.


-----

We are glad to answer your further questions on our submission.


Regards,

Authors

---

### Decision · Program_Chairs · 2024-09-25

**Decision:**

Reject

**Comment:**

The paper investigates how Graph Convolutional Networks smooth node features. The authors show that adjusting the projection can modify the normalized smoothness to any desired level. The paper further introduces a new method, SCT, which allows GCNs to learn node features with a targeted smoothness level, thereby improving node classification.

Reviewers generally agree that this is a borderline paper, with both reasons to accept and reject it. They appreciated the new insights into node smoothness in GCNs and noted that SCT enhanced accuracy across various architectures and datasets, particularly highlighting the significant performance improvement when applying SCT to heterophilic datasets.

However, reviewers also expressed serious concerns about the lack of solid experimental evidence, pointing out that important baselines appear to be missing. They suggested that the paper's concepts should be more thoroughly connected to existing literature. Additionally, they noted that the analysis is relatively narrow, focusing mainly on GCN and ReLU/Leaky ReLU, and that the results are not surprising given the context provided by [4]. They also pointed out that the discussion of homophilic and heterophilic settings is missing. Specifically, since there is a gap between the theory presented (which currently analyzes generic graphs without distinguishing between homophilic and heterophilic settings) and its application, the reviewers recommend including both an ablation study and a discussion addressing this distinction.

For these reasons, I recommend rejecting the paper.